# Fully 3D printed flexible, conformal and multi-directional tactile sensor with integrated biomimetic and auxetic structure

Yuyang Wei [1,2], Bingqian Li [3], Marco Domingos[1], Zhihui Qian[3], Yiming Zhu [1], Lingyun Yan[1], Lei Ren[1,3✉] & Guowu Wei [4✉]

Tactile sensors play a crucial role in the development of biologically inspired robotic prostheses, particularly in providing tactile feedback. However, existing sensing technology still falls short in terms of sensitivity under high pressure and adaptability to uneven working surfaces. Furthermore, the fabrication of tactile sensors often requires complex and expensive manufacturing processes, limiting their widespread application. Here we develop a conformal tactile sensor with improved sensing performance fabricated using an in-house 3D printing system. Our sensor detects shear stimuli through the integration of an auxetic structure and interlocking features. The design enables an extended sensing range (from 0.1 to 0.26 MPa) and provides sensitivity in both normal and shear directions, with values of 0.63 KPa$^{-1}$ and 0.92 N$^{-1}$, respectively. Additionally, the sensor is capable of detecting temperature variations within the range of 40−90 °C. To showcase the feasibility of our approach, we have printed the tactile sensor directly onto the fingertip of an anthropomorphic robotic hand, the proximal femur head, and lumbar vertebra. The results demonstrate the potential for achieving sensorimotor control and temperature sensing in artificial upper limbs, and allowing the monitoring of bone-on-bone load.

[1] Department of Mechanical, Aerospace and Civil Engineering, School of Engineering, The University of Manchester, Manchester M13 9PY, UK. [2] Department of Engineering Science, University of Oxford, Oxford OX1 3PJ, UK. [3] Key Laboratory of Bionic Engineering, Ministry of Education, Jilin University, Changchun, China. [4] School of Science, Engineering and Environment, University of Salford, Manchester M5 4BR, UK. ✉email: lei.ren@manchester.ac.uk; g.wei@salford.ac.uk

Over the past decade, great attention has been dedicated to the development of wearable, flexible tactile sensors and electronic skin for application in prosthetics, robotics and other healthcare devices[1–13]. These applications often require the use of tactile sensors able to detect shear stimuli. To that end, different strategies have been proposed, including the use of advanced composite materials, multi-layered biomimetic or hierarchical structures[3,9,13–15]. However, the fabrication of high-performance sensors with conventional manufacturing techniques such as molding, photolithography and etching[16–19] remain challenging, mainly due to the complex and time-consuming fabrication processes and material preparation stages. In terms of design, sensors often display simple, planar geometries which are incompatible with the majority of working surfaces (irregular or uneven), thus limiting their performance and range of applications[20–22]. Soft optical[23,24] and capacitive sensors[25–27] have also been proposed by other research groups and integrated into robots to achieve sensorimotor control. However, the use of such sensors remains hindered by the large space required to accommodate the camera and other supporting components. Additionally, capacitive sensors can be affected by temperature and humidity, which can lead to inaccurate readings.

In an attempt to improve the sensitivity and linear sensing range of tactile sensors, several groups have reported the combination of highly conductive composite materials such as metallic particles, nanowires[9], carbon nanotubes (CNT)[28] and graphene palate[29] with an insulation matrix. Carbon nanotubes CNT/polymer composites displaying high axial conductivity are particularly attractive for the generation of sensors with dielectric layers. When compared to similar quantities of carbon black or graphene, the addition of CNT to polymer matrices induces a substantiality higher piezoresistive response with minimal effect on the mechanical properties of the composite[30–32]. Other approaches based on volatile or water soluble micro-particles have also been employed to generate sponges or porous material[11,29] with higher self-contact area and larger linear sensing range. However, the process for producing these porous materials is time-consuming and technically complicated. Also, the stability and homogeneity of the material cannot be precisely controlled thus negatively affecting the final structure and performance of the sensor. Alternative routes toward the design of sensors with improved sensitivity and linear sensing range include the use of multiscale hierarchical structures[8,9,33], biomimetic interlocked structures[3,34–36], and mechanosensory hair or even crack-like structures[37,38]. The integration of microspheres with nanomaterials has also been used as a potential avenue to obtain hierarchical structures with enhanced piezoresistive properties[14]. In recent studies[3,13,33], a series of interlocked microdome arrays emulating intermediate ridges in the human skin have been proposed as ideal systems to amplify signal responses and differentiate between multiple stimuli directions. Because of their biomimetic structure these systems also found application as optical-based tactile sensors to improve the capability of discriminating fine features[24]. In a similar vein but using multilayer or hierarchical hair-like interlocked geometries[9,12], other researchers have demonstrated the ability to produce structures with improved sensitivity and extra low detection threshold of 0.6 Pa. However, these structural features require the use of sophisticated and time-consuming fabrication processes. The pressure sensor that can be 3D printed or fast-prototyped was developed by Guo et al.[39]. while a simple coil structure was adopted and limit the sensing performance. The expensive Ag-silicone rubber material was used for printing, which limit its practical applications in industry. The sensing area of this 3D printed was also small, less than 10 mm². Despite notable progress, the reality is that most of the sensors reported above have not yet found a practical application in the industry. The extrusion-based printing process was adopted for this research due to its flexibility in material selection, high throughput and better mechanical properties compared to the other conformal printing process such as the inkjet printing and aerosol jet printing[40–45]. To expedite the translation of this technology from the laboratory to practical applications and usher in the next generation of prosthetics, it is imperative to develop the strategies for fabricating pressure sensors that possess the following characteristics: (1) employ less complex and more cost-effective manufacturing processes; (2) exhibit comparable or superior sensitivity and linear sensing range compared to existing sensors; (3) establish a stronger, more stable, and durable interface with the working surface; (4) offer customized sensing areas ranging from millimeters to centimeters in size. In this paper, we introduce a flexible tactile sensor capable of detecting both contact pressure and environmental temperature. Notably, we integrate a biomimetic and auxetic structure into the sensor, enabling the detection of shear stimuli. Leveraging 3D printing technology, we demonstrate the feasibility of our approach to rapidly produce low-cost conformal sensors directly on uneven working surfaces. Moreover, these sensors can be flexed to conform to different surfaces. Through experimentation, we showcase the effective utilization of in-situ printed sensors for achieving sensorimotor control and temperature sensing capabilities in a biomimetic hand. Additionally, we validate their accuracy in monitoring bone-on-bone loads in human joints, such as vertebrae and femurs.

When compared to sensors with a planar structure in this study and others in the literature[11,46–49], our unique sensor design with integrated biomimetic interlocked and auxetic structure provides benefits in terms of sensing performance, namely: (1) a larger linear sensing range due to the negative Poisson's ratio; (2) higher sensitivity at low pressures (10 KPa); and (3) easy detection of the stimuli direction. Besides the outstanding pressure sensing performance, the sensor can also respond to temperature variations between 40 and 90 °C with a sensitivity of 0.27% °C$^{-1}$. More than simply overcoming the limitations of the current technology, this study aims to pave the way towards the design and fabrication of the next generation of tactile sensors with user-defined auxetic features suitable for applications in robotic/prosthetic hands and in pressure/temperature monitoring in impaired human joints.

## Results

**The tactile sensor—design and 3D printing**. The layout of the flexible tactile sensor array consisting of 121 sensing elements is depicted in Fig. 1a−e. The structure contains the upper and lower papilla-auxetic sensing layers and is sandwiched between two flexible electrode layers. The sensing area and distribution density of the tactile elements are comparable with slowly adapting type I and fast adapting type I mechanoreceptor to ensure a close match between the sensing capability of the proposed sensor and that of a human subject[50]. The resistive-type sensor consists of two sensing layers, and two electrodes, which are sandwiched together. When an external pressure is applied, the sensing layer is compressed, resulting in a change in the electrical conductivity of the composite. The change in conductivity is detected by the electrodes and can then be converted into a corresponding pressure signal. The biomimetic interlock structure is integrated to enhance the sensitivity and provide capability of discriminating between different directions of the external stimuli. Our results show that the piezoresistive sensing elements (located at the position of each small papilla) placed on the side of the stimuli experience a substantially larger reduction of resistance compared

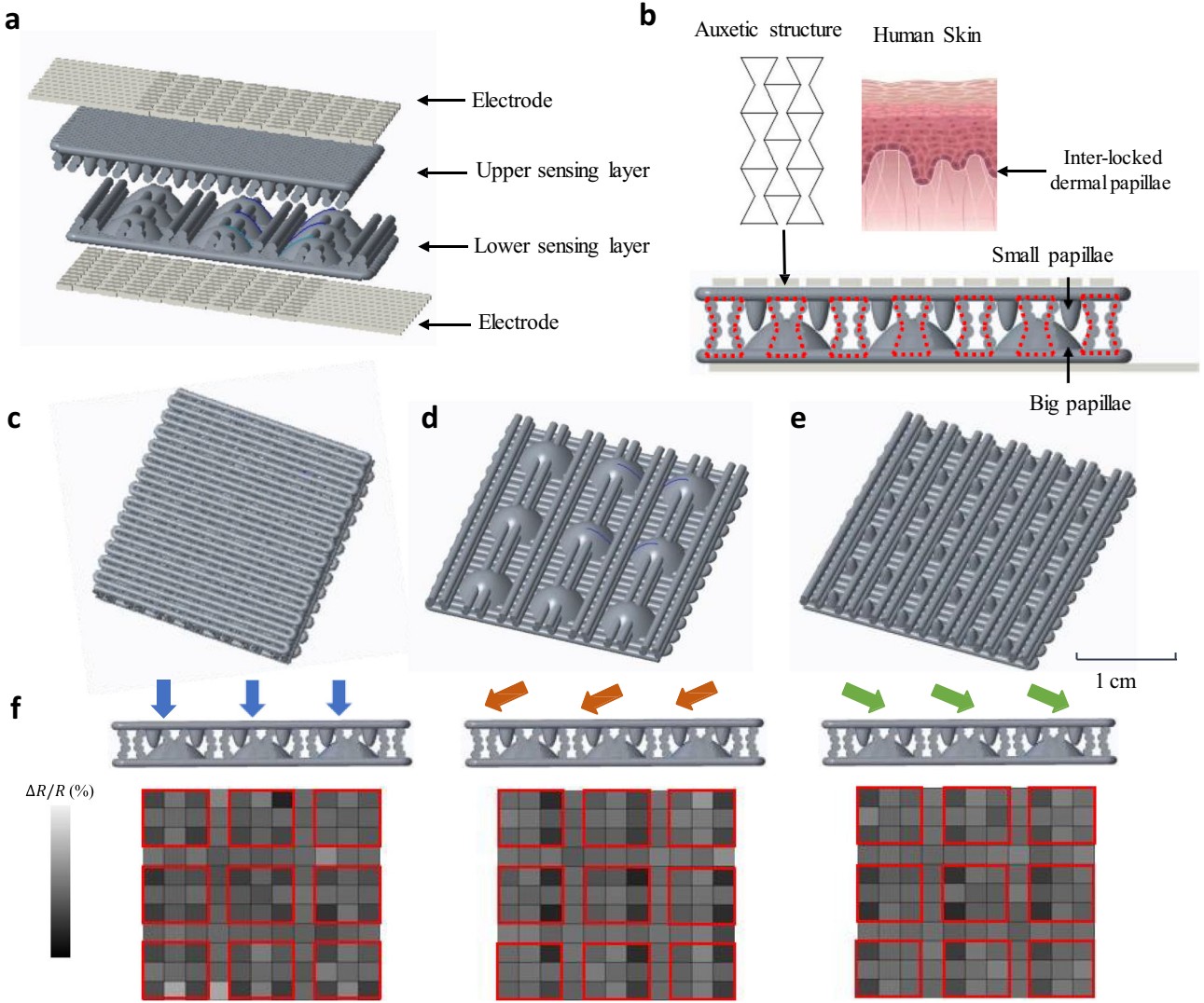

**Fig. 1 The structure and working principle of the tactile sensor. a** A standard 11 × 11 tactile sensor array is used, where each sensing element corresponds to a small papilla of the interlocked structure. **b** The sensor features an integrated auxetic and biomimetic interlocked structure, with the auxetic structure highlighted in red dash lines. The small and large papilla are located on the upper and lower sensing layers, respectively. **c** The 3D model of the sensor. **d** The lower sensing layer of the sensor consists of nine large papilla structures and auxetic lines. **e** The upper layer of the sensor consists of 36 small papilla structures and auxetic lines. **f** The stimuli are applied from different directions (indicated by the arrows above the sensor) and the corresponding pressure distribution shown through a customized graphical user-interface (GUI). The direction of the stimuli can be intuitively differentiated and classified into four principal directions based on each large papilla (highlighted with red squares) surrounded by four small papillae above its four corners. Therefore, the resolution of the shear force differentiation is 90° for this sensor. The sensor is based on the piezoresistive effect, where the change in resistance of the material is proportional to the applied pressure. The resistive-type sensor consists of two sensing layers (CNT/graphene/silicone composite) and two electrodes (silver-coated copper/silicone composite), which are sandwiched together. When an external pressure is applied, the sensing layer is compressed, resulting in a change in the electrical conductivity of the composite. The change in conductivity is detected by the electrodes and can be converted into a corresponding pressure signal.

to those placed on the opposite side. This can be associated with the geometric features of the lower large papilla and the anisotropic deformation of the upper sensing layer of the small papilla. The direction of the stimuli can be intuitively differentiated based on each large papilla surrounded by four small papillae above its four corners, as highlighted in Fig. 1f (red squares). The two small papillae on the side of the stimuli experience larger current increments than the other two on the opposite side. Therefore, the pressure mapping on the tactile sensor array, originating from the anisotropic distribution of resistance around the lower papilla, provided the ability to differentiate between the different directions of the stimuli. On the other hand, the single sensing element was not able to provide bulk information on the directions of the

external stimuli. A graphene/CNT/silicone rubber composite is used as the printing material for fabricating the proposed tactile sensor. The graphene platelet and multi-wall CNT are evenly distributed within the silicone matrix (Fig. 2a), providing piezoresistive and thermosensitive properties to the sensor. The CNT to graphene weight ratio was optimized to maximize the temperature resistive coefficient. The conductive micro-copper-silicone composite is employed to print the electrode. Printability of the developed composite materials is determined through oscillatory rheological measurements, particularly by measuring the variation of viscosity and shear stress as a function of shear rate (see Supplementary Figs. S1 and 2). Both composite materials exhibit a shear-thinning behavior typical of

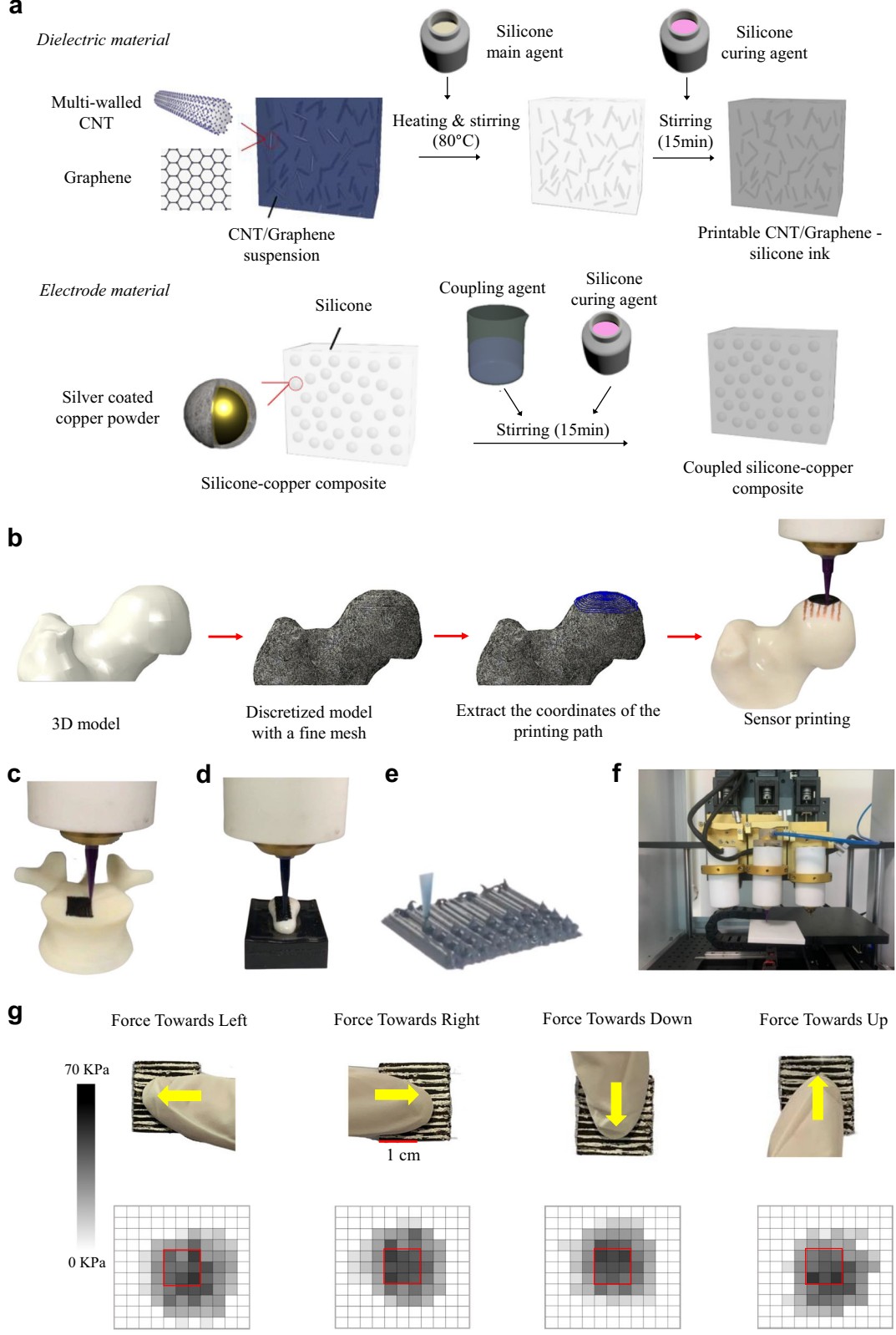

non-Newtonian systems, with extrusion being enabled by a reduction in viscosity as a consequence of increased shear stresses.

The sensor is manufactured using a 3D printing platform developed in-house and based on the concept of Fused Deposition Modeling. The printing process of the sensor from the modeling stage to the conformal 3D printing onto the freeform surfaces is illustrated in Fig. 2b. Once the target surface

is identified and modeled, the object is discretized with a fine parametric mesh using the CAE software Abaqus (Dassault Systèmes Simulia Corp, Providence, RI). Then, the coordinates of the node close to the printing paths conformal with the desired surfaces are extracted. Finally, a G code containing the numerical coordinates of the nodes (i.e., printing pathway) is generated and sent to the printer, allowing the sensor to be directly fabricated

**Fig. 2 The material preparation and 3D printing of the sensor. a** Material preparation of the Graphene/CNT/Silicon rubber and micro-copper-silicone composite. A graphene/CNT/silicone rubber composite is used as the printing material for fabricating the proposed tactile sensor. The graphene platelet and multi-wall CNT are evenly distributed within the silicone matrix, providing piezoresistive and thermosensitive properties to the sensor. The conductive micro-copper-silicone composite is employed to print the electrode. **b** The printing process of the sensor on a model of the femur head involved discretizing the 3D model of the femur head with a fine mesh, after which the coordinates along the printing path were extracted for generating G code and printing the sensor. **c** The 3D printed sensor was placed on the human vertebra. The surface of the vertebra's plateau was discretized, and the printing path was programmed before the printing process. **d** The 3D printed sensor was also placed on the distal index phalange. The surface of the femur head's plateau was discretized, and the printing path was programmed before printing it. **e** The dot printing for inter-locked papilla structure. The printer is controlled manually during dot printing. **f** The in-house 3D printer. **g** The sensor's response when in contact with the fingertip is shown, with the direction of the shear stimuli indicated by the yellow arrows. The distribution of the contact pressure initiated by the external stimuli is shown through a customized graphical user interface. It can be observed that the sensing elements on the sides of the external stimuli experienced a larger resistance variation. The directions of the shear force can be roughly identified and classified into four principal directions as is shown here, so the resolution of the shear force differentiation is 90° for this sensor.

onto the working surfaces (see Fig. 2c−f). The upper and lower layers of the sensor are printed separately. The former is printed onto the negative surface' and peeled off to attach with the lower layer on the working surface. The negative surface is modeled conformally to the positive working surface and then printed through Stereolithography (SLA). The G codes for printing the sensors onto different positive working surfaces and negative surfaces are presented in the supplementary material.

Figure 2g shows the response of the sensor under contact with the fingertip. In the trials, the fingertip pressed the tactile sensor, and the direction of the shear stimuli was indicated by the yellow arrows. The distribution of the contact pressure initiated by the external stimuli is shown through a customized graphical user interface (GUI). It can be seen that the sensing elements on the sides of the external stimuli experienced a larger resistance variation.

**Auxetic structure optimization of the tactile sensor**. Previous research has shown that auxetic structures could provide negative Poisson's ratio and larger self-contact area[51]. To enhance the sensitivity of our sensor in terms of contact sensing and enlarge the linear sensing range, we have chosen to use an auxetic structure in this work. The top row in Fig. 3a shows the 3D models of the sensors, while the bottom row shows the corresponding physical prototypes used for structure optimization. The thickness and re-entrant angle of the auxetic structure (as shown in Fig. 3b) were optimized to achieve the best nominal sensitivity. The simulation results were validated against experiment data, and a detailed description of the structure optimization is presented in the 'Method' section.

The simulation results showed that the sensor with a re-entrant angle of 65° and an H/L ratio of 1.60 achieved the largest nominal sensitivity. Figure 3b presents the optimization results and the associated experimental validation. The transparent mesh represents the real sensor characterization results, while the solid mesh depicts the simulation results. Both the experimental and simulation results suggest that the sensitivity and linear sensing range are enhanced by the integrated inter-locked and auxetic structure. To validate the FE simulation results, twenty physical tactile sensors were fabricated, with re-entrant angles of 45°, 60°, 75°, 90°, and five H/L ratios. The enlarged diagram of the fabricated sensor with the re-entrant angle of 45º is presented in Fig. 3c, the sensors under compression pressure of 50 and 100 KPa were shown in Fig. 3d. The simulation output of compression tests for sensors with different auxetic features is presented in Fig. 3e, Fig. 3, and Supplementary Movie 1 in the supplementary material.

**Characterization of the sensor for pressure sensing**. Figure 4 illustrates the characterization of the proposed tactile sensor,

including sensitivity and linear sensing range (Fig. 4a−c). Six sample sensors with the same size for each of these three different structures (planar, inter-locked structure, and the auxetic with inter-locked structure) were tested and the similar performances were observed. Six sample sensors with the same size for each of the three different structures (planar, inter-locked structure, and auxetic with inter-locked structure) were tested, and similar performances were observed. The sensitivity of the planar sensor (under 10 KPa of pressure, which is the threshold pressure of human touch applied during routine activities[52]) was below $0.03\,\mathrm{KPa}^{-1}$ (±0.01) but it increased to $0.63\,\mathrm{KPa}^{-1}$ (±0.04) by integrating the interlocked papillae and optimized auxetic structure. Additionally, using the proposed structure, the linear sensing range increased from 0.1 (±0.02) to 0.26 (±0.03) MPa, with a high correlation coefficient of 0.95., which is higher compared to the values displayed by most of the published high-performance sensors[2,11,28,46–49]. The shear sensitivity of the proposed tactile sensor is evaluated under different normal contact pressures ranging from 0.2 KPa to 10 KPa, using a customized horizontal testing platform with a push-pull dynamometer (see Supplementary Fig. S4) for generating shear force (up to 5 N) and a universal testing machine for normal compression. The inter-locked features provide the sensor with the ability to differentiate between the directions of external stimuli and improve its sensitivity. Furthermore, the integration of the auxetic structure enhances the sensitivity and extends the linear sensing range. The maximum shear sensitivity of $0.92\,\mathrm{N}^{-1}$ (±0.08) is found under the normal pressure of 0.2 KPa and seems to be reduced with the increased contact pressure induced by the normal compression. The shear sensitivities are above $0.39\,\mathrm{N}^{-1}$(±0.06) under compression pressures ranging from 0.4 to 0.8 KPa and are reduced to $0.24\,\mathrm{N}^{-1}$ (±0.04) when the normal pressure is above 1 KPa. It can be found from Fig. 4c that the shear sensitivity is improved due to the inter-locked and auxetic structure of the sensor. The sensitivity of the planar layer is found to be $0.22\,\mathrm{N}^{-1}$ (±0.05) but with the integration of the biomimetic and auxetic features this value increased to $0.92\,\mathrm{N}^{-1}$(±0.08). The lower detection limit is approximately 50 Pa as shown in Fig. 4d, and the response time is approximately 10 ms (±3.00) as recorded by the multimeter. In addition, the tactile sensor is tested under repeated pressing and releasing cycles in a wide range of compressive deformation. A stable signal response was achieved after subjecting the sensor to 1500 loading-unloading cycles with a pressure of 100 KPa, as shown in Fig. 4e. Additionally, the sensor displayed good durability, as evidenced by the signal response during the last 50 cycles of compression. The hysteresis was also measured based on the first and last 50 cycles and found to be 8.2% ± 1.7%. The diagram presenting the hysteresis loops of sensor at various scanning rates up to 100 kPa (see Fig. S5 in the supplementary material). To evaluate the dynamic accuracy of the sensor, it was attached to a vibration platform (HTA-3000A, Huitai Ltd.,

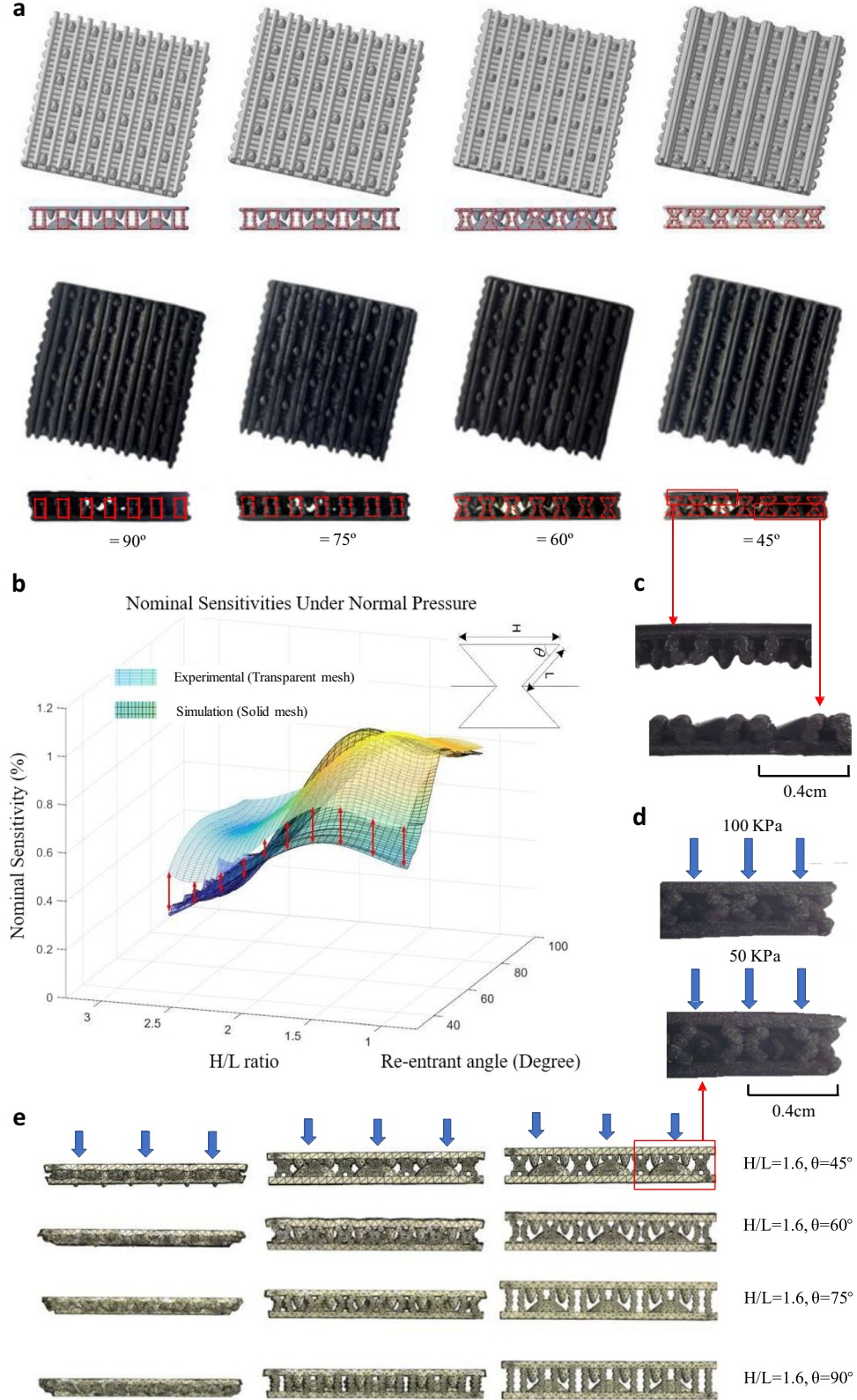

**Fig. 3 The optimization results of the sensor and the prototypes with different structures. a** The fabricated sensors and the corresponding 3D models. The sensor with the re-entrant angles of 45°, 60°, 75°, 90° are presented. The auxetic structures with different re-entrant angles are highlighted with red dash lines on 3D models. **b** The optimization results for the auxetic structure. The largest nominal sensitivity is achieved under the re-entrant angle of 65° and H/L ratio of 1.6. **c** The enlarged diagram of the fabricated sensor with the re-entrant angle of 45°. **d** The fabricated sensor with re-entrant angle of 45° under the compression pressure of 50 and 100 KPa. **e** The simulation results of the compression experiment for the sensor with the re-entrant angle of 60° and the H/L value of 1.6. More simulation results of the sensors with different re-entrant angles under compression are presented in Supplementary Fig. S3.

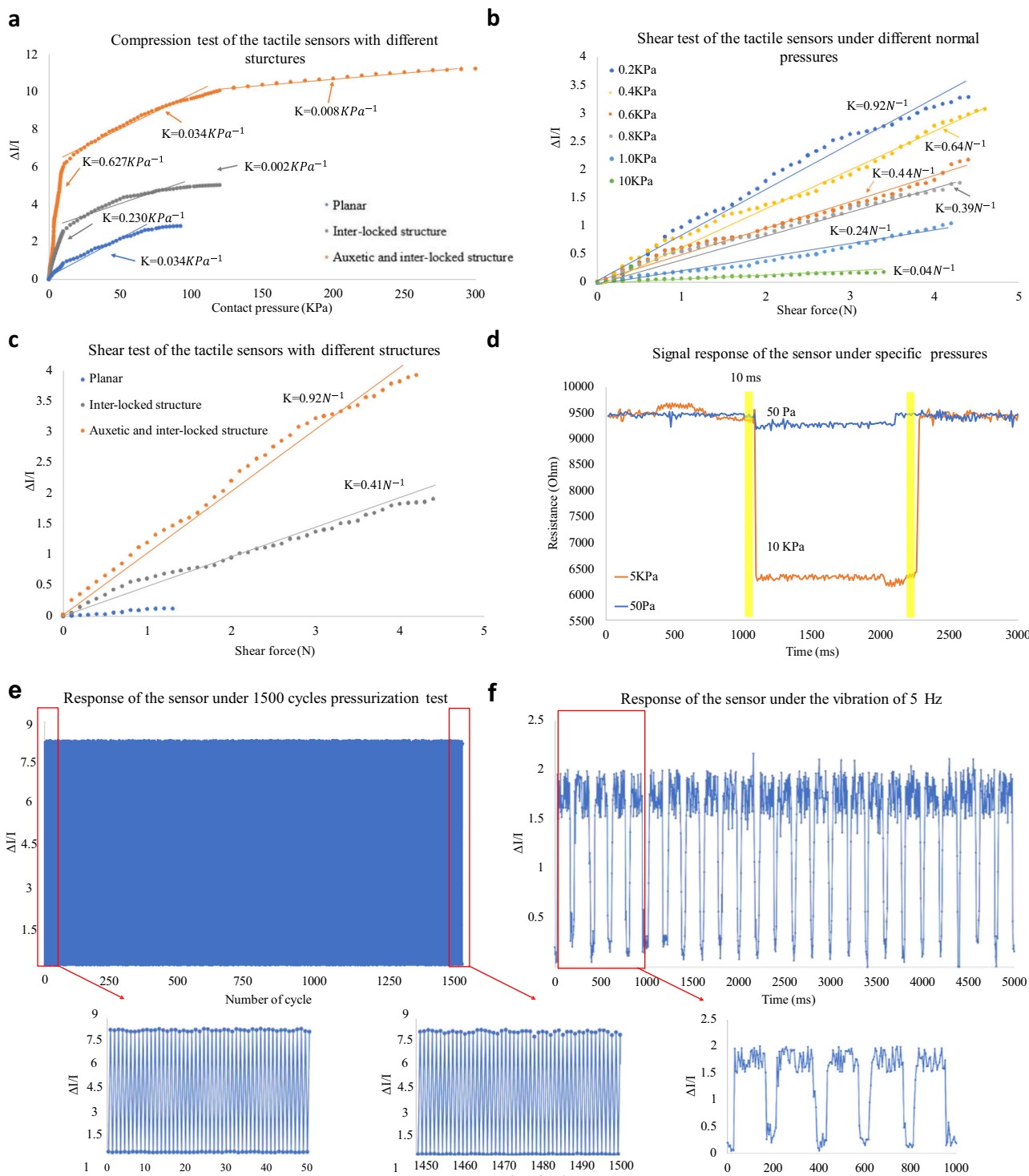

**Fig. 4 The characterization of the tactile sensor in terms of pressure sensing and a comparison with other published pressure sensors are presented. The enhancement of the pressure sensing performance after integrating the biomimetic and auxetic features is also demonstrated. a** The sensitivity of the sensors with different structures, including the planar sensor, the sensor with the inter-locked structure, and the sensor with both inter-locked and auxetic structures, was evaluated and compared. The three different structures of the sensor are presented in Fig. S8 in the supplementary material. **b** The shear sensitivity of the sensor under different normal pressures varied from 0.2 to 10 KPa. **c** The signal response of the sensor under the lowest detection limit 50 Pa and gentle touch pressure 10 KPa. **d** The shear sensitivity of the sensor with three different structures. **e** The response of the sensor under 1500 cycles pressurization test, the start and final 50 cycles are enlarged. **f** The response of the sensor under the vibration of 5 Hz.

China), and a standard weight was added to the surface of the sensor to produce stimuli. The vibration frequency was varied under 5 Hz, and the resulting signal output of the sensor was displayed in Fig. 4f. The results indicate that the sensor was able to accurately track the vibrating signal, with the same frequency of pressure variation observed. The dynamic accuracy of the sensor was then calculated for the three different vibration frequencies based on the mean square error, assuming that the

stimuli signal from the vibration platform was an impulse signal. The experimental findings suggest that the dynamic accuracy of the sensor is approximately 0.82 KPa within a frequency range of 5−20 Hz. The static accuracy of the sensor is 1.67% under a pressure of 0.25 MPa. The sensing performance of the published sensors[11,46–49] and the sensor introduced in this study are compared in Fig. 5a. The effects of the biomimetic inter-locked and auxetic structure on sensor performance are presented in Fig. 5b. The signal-to-noise ratio (SNR) was also evaluated using the standard sample sensor and the experimental results are shown in Supplementary Fig. S6. The SNR is approximately 18.7 dB which is higher/comparable to those piezoresistive pressure sensors in the literature[53–56] due to the optimized material composition and structure design.

**Monitoring magnitude and direction of the bone-on-bone load through the senor**. To demonstrate the sensor array's ability to detect both the magnitude and direction of external stimuli, the researchers printed the sensor directly onto the plateau of a human vertebra model and used it to monitor bone-on-bone contact. Physical models of the L2 and L3 vertebrae, along with the intervertebral disc (shown in Fig. 6a−c), were 3D printed with SLA and used as the working surfaces of the sensor. Negative surfaces for printing the upper layer of the sensor were also created with SLA. The printed sensor was wired with a customized electric circuit for collecting the analog outputs from the sensing elements. The jumper wires were connected to the sensor using the same material that was used for printing the electrode, and the tip of the metal wire was stuck to the electrode during the solidification of the composite material. The sensor fabrication process from 3D design to the final stage of sensor testing is shown in Fig. S7 of the supplementary material. These analog signals were then converted into digital signals and visualized on the GUI interface, as shown in Fig. 6d. In this experiment, the spine's movements, including lateral bending, axial rotation, and flexion/extension, were performed under different compression forces ranging from 25 to 100 N. The GUI interface displayed the pressure generated by the corresponding contact force acting on each tactile element in gray color, with darker shades corresponding to higher values. The pressure tended to be more

intense on the side being crushed during flexion and lateral bending. In this case, the average contact pressure increased from approximately 17 KPa to 65 KPa under the compressive forces ranging from 25 to 100 N. The maximum contact pressure of 70 KPa was observed at the center of the lower vertebra due to the concentrated external load. The results showed that the sensor printed directly onto the vertebra performed well, as expected, under arbitrary contact between the intervertebral disc and the vertebra.

The proposed tactile sensor is also printed onto a model of a proximal femoral bone to monitor the bone-on-bone load of the human hip joint (see Fig. 6e). A conformal printing path around the femoral head is initially defined to print the sensor. The proximal femoral head and part of the pelvis are printed to perform the motions of the hip joint during the swing of the lower limb. The test results are illustrated in the GUI interface and show a reasonable pressure distribution during the whole swing phase. Therefore, the experimental results reported above suggest that this 3D printed tactile sensor can be rapidly fabricated onto geometrically complex surfaces such as the vertebra or femoral head and used for monitoring the bone-on-bone contact. The printed sensors working on the hip joint and between the lumbar vertebrae are shown in Supplementary Movie 1 and the G codes for printing the sensors are presented in Supplementary Data 1−4.

**Sensorimotor control of the biomimetic hand through the tactile sensor**. The dimensions and shape of the tactile sensor can be customized according to the sensing area, which ranges from millimeter to centimeter scale, and the working surfaces, respectively. To demonstrate the scalability and adaptability of the sensor, an $18 \times 10$ mm$^2$ tactile sensor is printed onto the distal phalange of the index finger in a biomimetic anthropomorphic robotic hand, as shown in Fig. 7a−c. The tendon-driven biomimetic hand used in this study is reconstructed from a male subject, including the intact bone skeleton, tendons, interphalangeal ligaments, and skin. Electric motors actuate the anthropomorphic robotic hand and mimic the human-like kinematics and grasping quality. The distal phalange of the index finger is disassembled from the biomimetic hand to allow

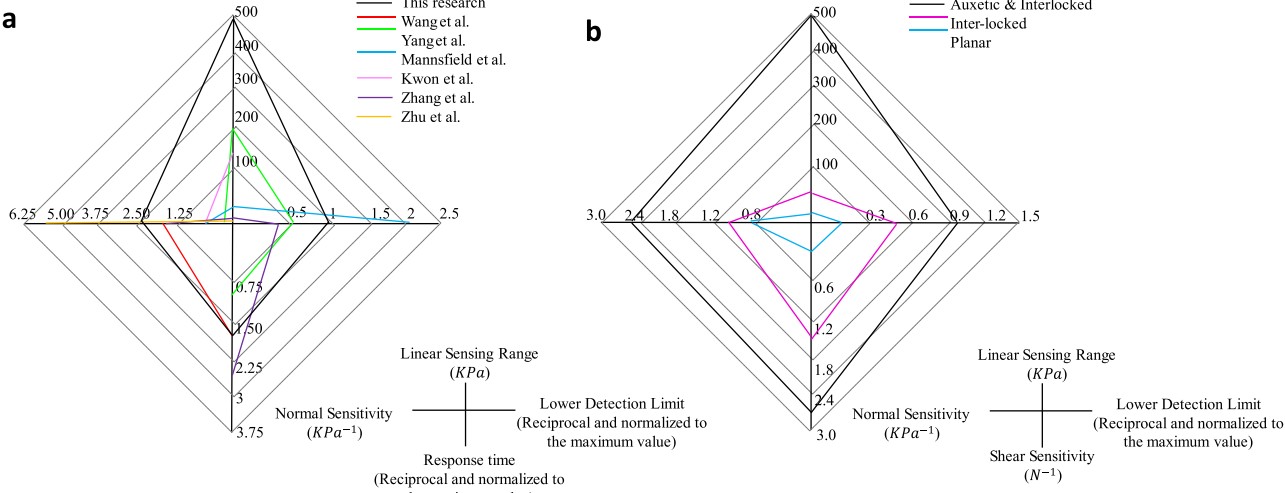

**Fig. 5 Comparisons of sensing performance between the tactile sensor developed in this research with other published sensors and also among the different tactile sensors developed in this study. a** The comparisons of sensing performance between the tactile sensor developed in this research with other published sensors. The lower detection limits and response times are reciprocal and then normalized to the maximum value, so a larger index of the lower detection limit in the diagram indicates better sensor performance. **b** The comparisons of sensing performances among the planar sensor, the sensor only adopting interlocked features, and the one with the integrated inter-locked and auxetic structure.

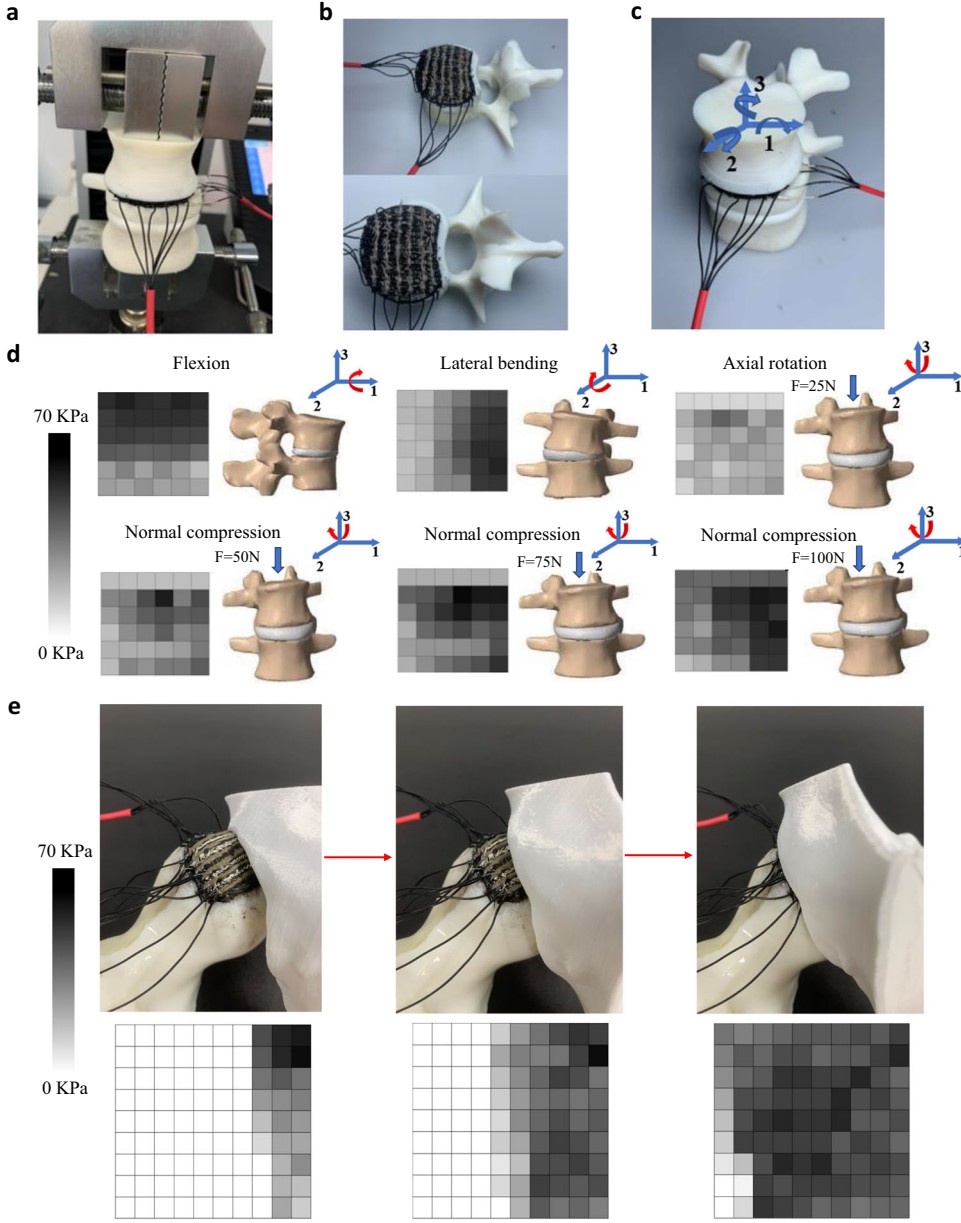

**Fig. 6 The applications of the sensor for monitoring bone-on-bone load. a** The sensor is printed onto the vertebra plateau for monitoring contact pressure. **b**, **c** The printed sensor on the vertebra plateau and the rotation around three axes corresponding to flexion/extension (axis 1), lateral bending (axis 2) and axial rotation (axis 3). **d** The signal distribution of the sensor under motions of the spine vertebra, including flexion, lateral bending, and axial rotation. The axial rotation is combined with different compression forces of 25, 50, 75, 100 N to show the sensor's performance under different combinations of loading conditions. **e** The printed sensor on the distal femur bone of the human hip joint model and the signal distribution of the sensor during the swing of hip joint.

for printing the tactile sensor. A customized electric circuit is integrated to collect the analog outputs from the tactile sensing elements, and a Python-based controlling algorithm is developed to convert sensor analog outputs to digital signals, providing the robotic hand with tactile feedback for grasping performance. Detailed information about the hardware and control of the robotic hand is presented in the "Methods" section.

Using the setup described above, it was possible to create a realistic scenario where the grasping action of the biomimetic hand is disturbed by an external force, and the response of the tactile sensor is measured in terms of feedback. For this purpose, a pulling force of 0.75 N was directly applied to the ball, which counteracted the grasping movement of the hand. Figure 7d, e show the reaction control of the biomimetic hand with and

without the tactile feedback, respectively. The experimental results demonstrate that the tactile sensor is able to perceive the shear stimuli caused by the pulled object (i.e., ball) and initiate the sensorimotor control to maintain stable grasp. This effect is not visible in the absence of tactile feedback, and the external disturbance could easily break the grasping stability. The Python code controlling the kinematics of the bionic hand is provided in Supplementary Data 5.

From the above applications of the proposed tactile sensors in bone-on-bone load detection and in robotic finger tactile sensing, it can be seen that the proposed tactile sensor can be rapidly integrated, i.e., rapidly printed, onto a simulated biological system for monitoring the magnitude and direction of stimuli/load/force, and providing tactile feedback for sensorimotor control.

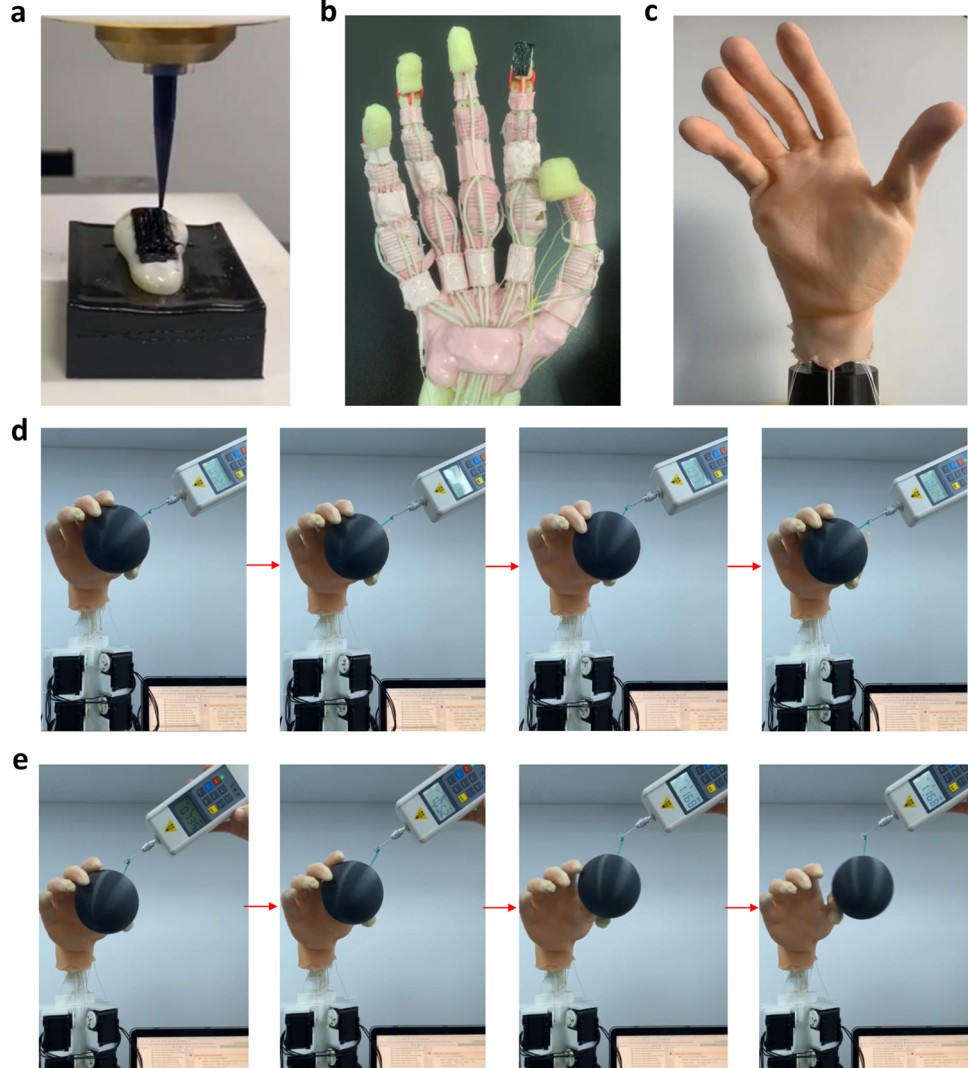

**Fig. 7 The grasping performance of a biomimetic hand with tactile sensor printed on the index fingertip. a** The tactile sensor (indicated in red circle) is printed directly onto the distal phalange of index finger. **b** The distal index finger bone with sensor is assembled back to the robotic hand. **c** The robotic hand is then covered by the artificial skin. **d** A plastic ball is grasped by the biomimetic hand, approximately 0.75 N pulling force is applied onto the ball. The biomimetic hand with tactile feedback from the sensor can sense the shear stimuli caused by the pulled object and maintain a stable grasp. **e** The biomimetic hand without tactile feedback cannot produce a stable grasp.

**Characterization of the sensor for temperature sensing and the application on robotics**. The temperature sensing capability is assessed by placing the sensor inside a heating chamber (see Supplementary Fig. S9) developed in house. The temperature variation and the signal response of the sensor are recorded to compute the temperature coefficient of resistance (TCR). TCR is a key parameter for evaluating the sensing performance of the temperature sensor and is defined as:

$$S = (\Delta R / R_0) / \Delta T \qquad (1)$$

where $\Delta R$ is the change of the resistance and $R_0$ is the resistance measured at room temperature. $\Delta T$ is the change of the applied temperature.

The weight ratio of graphene to CNT was optimized to achieve the most stable and highest TCR, and the detailed experimental setup and material optimization are presented in the Method section. In Fig. 8a, the temperature response of the sensor under five thermal cycles is shown, and a stable and linear relationship between the relative resistive change and temperature variation with a sensitivity of 0.27% °C$^{-1}$ is achieved. The temperature

sensor has a sensing range from 40 to 90 °C. The response of the sensor under specific temperatures of 50 and 90 °C is presented in Fig. 8b, where the devices are heated from room temperature to 50 or 90 °C and maintained at these peak temperatures for approximately 1 min before cooling down to room temperature. The sensor can respond to the temperature variation with a reasonable response time, and the variation of the relative resistance change follows well with the heating profile. In Fig. 8c, d, it is demonstrated that the tactile sensor printed on the fingertip of the biomimetic hand could sense the temperature variation of a glass with hot water and provide feedback to release the firm grasping. The resistance of the sensor varies with changes in both contact pressure and temperature. It is important to note that the two different signals cannot be effectively differentiated from one another. However, the reading of the pressure signals can be scaled under different temperatures to mitigate their effects or interference. The effects of temperature response under various contact pressures were studied and analyzed, as depicted in Supplementary Fig. S10. The temperature responses of the sensor were recorded under different contact pressures,

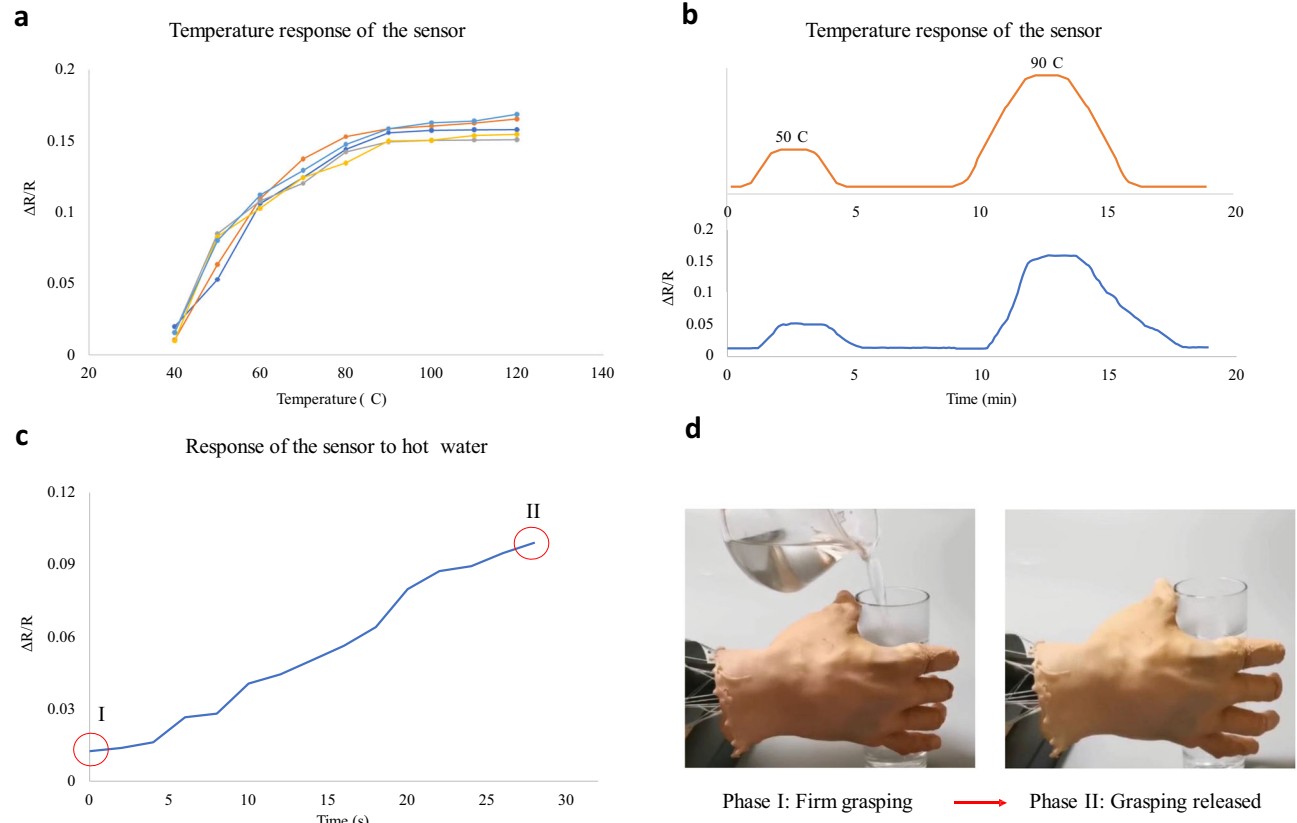

**Fig. 8 The characterization and performance of the tactile sensor in terms of temperature sensing. a** The temperature response of the sensor, five sensor sample were fabricated and tested. **b** The response of the sensor with the hot plate of 50 and 90 °C. The heating profile is shown on the top part of the figure while the relative change in resistance is plotted below. **c** The response of the sensor on the biomimetic hand to hot water was tested in two phases. In phase I, the biomimetic hand grasped the glass with room temperature water. In phase II, hot water was poured into the glass, causing an increase in temperature, and the sensor's response to the temperature change was measured. The grasp was then released. **d** The bionic hand under phase I and II.

illustrating the sensor's notable sensitivity to temperature variations. To account for different contact pressures, different scales were employed for temperature measurements, necessitating corresponding adjustments to the temperature readings. In Table S2, more detailed information on the sensor's piezoresistive properties and sensitivities under different pressures is provided to offer a clearer understanding of the pressure's impact on temperature sensing. Furthermore, our experimental results indicate that the impact of temperature on the sensor's piezoresistive properties is negligible when the temperature remains below normal room temperature (40 °C). The findings reveal that temperature variations ranging from 20 to 40 °C have minimal effects on the sensor's performance. As a result, in most cases, there is no need to scale or adjust the pressure readings when operating within the room temperature range. In summary, based on our experimental results presented in Supplementary Fig. S10 and Table S2, the temperature reading of our sensor can be scaled. However, for most scenarios, the pressure reading under room temperature does not require adjustment. These characterization results demonstrate the reasonable temperature sensing performance of the tactile sensor and its potential applications in the robotic hand.

## Discussion

Most of the tactile sensors developed over the past decade were manufactured using sophisticated technological processes such as molding[10,15,25,28,57], photolithography, and etching[16–19]. Although well established, these processes normally require highly skilled labor for the operation of the systems and preparation of materials, especially when advanced composite materials are used[9,12,15]. Also, previous research work has failed to properly address the need for compliance between the tactile sensor and the working surfaces, resulting in planar sensors with poor fitting to arbitrary surfaces[58,59]. The tactile sensor presented in this study offers advanced sensing capabilities for detecting tactile information, such as the direction and magnitude of stimuli, and can be efficiently and affordably fabricated to fit specific working surfaces. Furthermore, the sensor can respond to temperature changes with reasonable sensitivity. As presented in Fig. 8a, the linear sensing range of our sensor is larger than the pressure sensors reported in the literature[11,46–49]. The other key sensing indicators including normal and shear sensitivity and lower detection limit are also superior or at least comparable to those of previously reported sensors. additionally, the temperature sensing performance, including sensing range and sensitivity, is also comparable with carbon-polymer composite based[60–63] and commercial temperature sensors[64]. Unlike previous studies, we have utilized 3D printing technology to produce our sensor, resulting in shorter lead times and lower production costs when compared to previously published tactile sensors[13,16–19,33,35]. Further details on this comparison are provided in Table 1 of the supplementary material.

The inter-locked structure is employed in the sensor design to mimic the biomechanical characteristics of human skin, making the structure more sensitive to variations of strain components, and enabling the sensor to detect the direction of the stimuli.

Meanwhile, the auxetic structure is optimized and validated against experimental results. The re-entrant angle and thickness of the sensor are the main geometrical factors dominating the negative Poisson's ratio effect of the auxetic structure. These two factors are optimized, and the results suggest that a re-entrant angle of 65° and an H/L ratio of 1.60 can provide the best piezoresistive effect. The incorporation of the auxetic structure increases the self-contact area, resulting in a wider linear sensing range. Simulation results indicate that the negative Poisson's ratio enables a large self-contact area that linearly increases with applied forces in the sensor with biomimetic and auxetic features. The sensing range is significantly increased from 0.1 to 0.26 MPa. Since the piezoresistive characteristics of the elastic polymer-based sensor are dominated by stress/strain-related parameters during external stimuli, the higher sensitivity may have been caused by the sensitive SED variation to the external stimuli based on simulation results. The normal sensitivity is enhanced from $0.03\,\mathrm{KPa^{-1}}$ to $0.63\,\mathrm{KPa^{-1}}$ with the interlock and auxetic features, making the structure more responsive to normal pressure variations and leading to a better piezoresistive performance. The shear sensitivity is also enhanced (from $0.41\,\mathrm{N^{-1}}$ to $0.92\,\mathrm{N^{-1}}$ with inter-lock features), and the lowest detection limit can reach 50 Pa.

The sensitivity in terms of temperature sensing is not ideal (absolute value of TCR is below $0.0001\,°\mathrm{C^{-1}}$) based on the CNT/silicone rubber composite (see Supplementary, Fig. S11 (a)). Most of the twisted CNTs are stretched apart due to the thermodynamic expansion of the silicone rubber under increasing temperature, leading to more conductive pathways and reduced resistance. Therefore, the CNT/silicone rubber composite shows a negative TCR. However, the stretched CNTs cannot return to their original entangled state after cooling, resulting in a different distribution of conductive pathways and a decreased absolute value of TCR. To enhance thermosensitive stability and sensitivity, 2D graphene nanoplates are added together with CNTs into the silicone rubber. Due to the finely distributed nano plate structure, a relatively stable number of conductive pathways is achieved after a heating and cooling thermal cycle. The graphene plates spread out with the increased temperature, which dominates the decreased conductive pathways resulting from the thermal expansion of the silicone matrix. The experimental results also suggest that the temperature sensing performance, including sensitivity and stability, can be enhanced by adjusting the weight ratio between the two carbon nanomaterials (see Supplementary Figs S. 11b−d).

The sensor in this study has demonstrated that incorporating auxetic or other meta structures in the design can lead to improved performance. The optimized auxetic structure may offer insights for the design of tactile sensors in the future. Additionally, the tactile sensor can be printed directly and efficiently onto any uneven surface, such as the human phalange, proximal femur bones, and the plateau of lumber vertebra, ensuring perfect compliance with the working surfaces. Experimental and simulation results suggest that directly printing the tactile sensors onto curved surfaces results in negligible deformation effects on sensor sensitivity and linear sensing range, while other published tactile sensors still face critical issues with deformation effects. Moreover, the size and shape of the tactile sensor can be easily adjusted to accommodate different sensing area demands, ranging from $\mathrm{mm^2}$ to $\mathrm{cm^2}$. According to our experience using the in-house 3D printer, the smallest size of the printed sensor can be 3 mm by 3 mm, which is comparable to the smallest sensing element of commercial sensors such as TekScan 300E and Kitronyx MS 9723. The maximum surface area and curvature of the sensor depend on the limitations of the 3D printer. For our in-house printer, the maximum printing area is

$300*300\,\mathrm{mm^2}$, while the maximum curvature it can handle is approximately 2 mm. Due to the limited accuracy of the 3D printer, the minimum sensing element is $2\,\mathrm{mm^2}$, resulting in a maximum tactile density of around $16/\mathrm{cm^2}$. One limitation of our sensor is that the small uneven indenter could contribute to a similar pressure distribution as under shear forces. However, in most cases, the sensor is in contact with a surface without any bumps smaller than the sensing element. Although the small uneven indenter could contribute to a similar pressure distribution as under shear forces, this type of "uneven indenter" must have protrusions smaller than the sensing element, which is rare in real-life situations. To address this issue, a more sophisticated post-processing of the pressure signal could be developed using machine learning algorithms in future work. Also, the use of this sensor on tactile sensing actions of robotic hand, including realizing light contact with an unknown held object under sensorimotor control can also be completed in the nearly future which is a critical aspect to human-robot interaction and prosthetics development[65].

**Conclusion**. In summary, a fully 3D printed flexible tactile sensor based on biomimetic inter-lock and auxetic structure has been developed for sensing the contact pressures and the environmental temperature. Using a customized 3D printer, a Graphene/CNT/Silicone rubber and silver-coated powder-silicone composite was printed directly onto the working surfaces, fabricating the tactile sensors efficiently and economically. Similar performance to most of the published high-performance tactile sensors was achieved in terms of sensitivity, sensing range, and linearity[11,46–49]. The integration of biomimetic and auxetic structures is the key factor in achieving high sensitivity and a large linear sensing range for pressure and force sensing. The optimized weight ratio of CNT to graphene in the composite material ensures good temperature sensing performance. The use of the auxetic structure offers unique mechanical properties that can be effectively incorporated into the design of tactile sensors. The sensors can be rapidly fabricated onto various working surfaces, including the distal phalangeal bone, human vertebra, and distal femur bone, enabling monitoring of sophisticated biomechanical contact. The proposed tactile sensor also provides tactile and temperature feedback to the robotic hand, achieving excellent sensorimotor performance.

## Methods

**Material preparation for 3D printing**. The high purity (95 wt%) Multi-Walled CNT (XFM25, XFNano, China) and graphene nano platelets (XF021, XFNano, China) were uniformly dispersed in isopropanol by ultrasonic dispersion at room temperature for 30 min to obtain a CNT/Graphene suspension (see Fig. 2a). The silicone main agent (Polycraft GP3481-F Silicone rubber, MB Fiberfill, UK) was then added into the CNT/Graphene isopropanol suspension and heated to 80 °C under mechanical stirring until the isopropanol was completely evaporated. After cooling down to room temperature, the silicone curing agent was added to the composite in a weight ratio of 1:12. The graphene platelets were evenly dispersed in the silicone matrix and no obvious aggregation of CNT (see Supplementary, Fig. S12) was found based on micrographs. Our experimental results indicate that the CNT/silicone rubber composite with 7.5 wt% of CNT can achieve conductivity and a piezoresistive effect for pressure sensing (see Supplementary, Fig. S13), but the magnitude of TCR is below $0.1\%°\mathrm{C}{-1}$ and not stable after each heating cycle. To enhance the thermoresistive effect of the sensor, graphene nano plates are added, and the weight ratio of CNT to graphene is optimized to achieve the best temperature sensing performance

while maintaining a comparable piezoresistive effect with the CNT/silicone rubber composite. The conductivity of the CNT/silicone rubber composite is higher than that of the graphene/CNT/silicone rubber composite. To avoid a large effect on the piezoresistive effect and maintain a stable extrusion quality during printing, the total weight percentage of the carbon-based nano materials is kept constant at 7.5%, while the content of graphene plates is maintained below 4%. The thermal-resistive performance can only be achieved when the content of graphene plates is above 3.5% wt. Therefore, we fabricated tactile sensors with weight ratios of CNT to graphene of 4:3.5, 3.75:3.75, and 3.5:4, and characterized their temperature sensing performance (see Supplementary Figs. S11b−d). Our results show that the CNT/graphene/silicone rubber composite with 3.5 wt% of CNT and 4 wt% of graphene can achieve good stability and the highest TCR value among the sensors with different weight ratios of the two carbon nano materials.

Silver coated copper-powder-silicone composite (see Fig. 2a) was used to print the electrode of the sensor. It's prepared by mixing the silver coated copper palate (48 μm) with silicone rubber with the weight ratio of 1:3.5. The coupling agent (KH550) is then added under the weight ratio of 1:100 and mechanically mixed with the composite to improve the conductivity.

**3D printing platform and optimization of the printing configurations**. A customized 3D printing platform was used for the fabrication of the sensors. The system includes a 3-axis motion gantry with a workspace of $250 \times 250 \times 150$ mm³, linear motion converters from THK (Japan), two-stage gear reducers from Oriental Motor Ltd. (Japan), and servo motors from MiGe (China). A high-precision dispenser (S-SIGMA-X3-V5, Musashi Engineering, Japan) is integrated with a flexible Teflon tube with a diameter of 5.5 mm and a syringe as the material feeding device. The control system contains a 6-axis controller and a self-customized software. The 6-axis controller (Leadshine Technology Ltd., China) is used to process the G code from the software and control the motions of the feeding syringes and the 3-axis stage. This customized 3D printing platform has been used in our previous research for executing non-linearly varying material printing, and it has achieved high printing resolution and quality.

To achieve stable extrusion performance, the printing parameters such as printing speed and air pressure are optimized. The moving speed is varied from 5 to 15 mm/s (step increment of 1 mm/s) under pressure ranging from 0.3 to 0.7 MPa (step increment of 0.1 MPa) to carry out the trial line printing, resulting in a total of 121 different combinations of printing parameters. The pressure of 0.4 MPa with a printing speed of 12 mm/s is found to achieve stable printing quality and appropriate extrusion width. During the 3D printing process, in Step 1, a 15-gauge nozzle is mounted on the syringe loaded with silver coated copper-silicone composite material to print the lower electrode layer. Then, a 22-gauge nozzle mounted on another syringe with the Graphene/CNT/Silicon rubber composite is used for line printing the auxetic structure of the lower sensing layer. The dot printing is controlled manually to print the large papilla structure of the lower layer. The same printing parameters and patterns are applied to print the upper sensing layer, which is then attached to the lower layer. Finally, the upper electrode is printed on top of the sensor using a similar process as the lower electrode in Step 1. The 3D printing process of the sensor is shown in Supplementary Movie 1 in the supplementary material.

**Testing platform for characterization of pressure sensing**. In order to take full advantage of the piezoresistive and mechanical properties of the interlocked-auxetic features, the standard sensor

with the sensing area of $18 \times 20$ mm² is fabricated for the characterization of pressure sensing (see Fig. 4). The sensor prototype is powered by a 5 V DC supply and connected to a customized electronic circuit. The output current is measured and collected by a multimeter (Keysight 34465A, Keysight Ltd., HK) at a frequency of 500 Hz. The normal pressure applied to the sensor is controlled by a universal testing machine (WH-5000, Weiheng Co., China) for pressures above 10 kPa, and a precise push-pull dynamometer for pressures below 10 kPa. The push-pull dynamometer is mounted onto a horizontal tensile test platform (see Supplementary, Fig. S4) to produce shear force for characterizing the piezoresistive effects under stimuli from horizontal directions.

**Testing platform for characterizations of temperature sensing**. A custom-made heating chamber is developed in-house for providing a stable temperature variation during the characterization of the sensor. It is composed of a plastic box with dimensions of $75 \times 30 \times 15$ mm³ and a heating plate powered by a DC voltage supply. The variation in resistance is recorded by a multimeter (Keysight 34465 A, Keysight Ltd., HK) while the temperature is monitored using another multimeter (See Supplementary Fig. S9).

**The FE modeling and optimization of the sensor structure**. The tactile sensors with different auxetic structures are designed and modeled in Creo Parametric (PTC Inc. Boston, US). Two key geometric parameters including the re-entrant angle (from 45° to 90° with the increment of 1°) and the H/L (Fig. 3b) ratio (H/L = 2.7, 2.0, 1.6, 1.3, 1.0) are optimized. A total of 230 tactile sensors with different geometrical parameters are modeled for structural optimization (46 re-entrant angles together with 5 H/L ratios, resulting in $46 \times 5 = 230$ different auxetic structures). The compression test is simulated in the commercial FE software Abaqus. The heights of the interlocked large and small papilla structures are adjusted with the auxetic structure as $L \times \sin\theta$, while the diameters remain constant. The sensitivity in terms of the piezoresistive effect is defined as the objective of the structure optimization for this tactile sensor in the linear regime as:

$$S = (\Delta I/I_0)/\Delta P \quad (2)$$

where $\Delta I$ is the change of current flow over the sensing layers, $I_0$ is the base current measured without any external stimuli on the sensor, and $\Delta P$ is the change of the applied contact pressure.

Further, the nominal sensitivity, denoted as Sn, for optimizing the auxetic structure was defined as:

$$Sn = (\Delta SED/SED_0)/\Delta P \quad (3)$$

where $\Delta SED$ stands for the change of the strain energy density (SED) at the site of the sensing elements, and $SED_0$ is the SED computed under the maximum pressure of 0.26 MPa.

Research on the piezoresistive characteristics of elastic polymer-based sensors has shown that sensitivity is mainly influenced by the structure's geometry and stress/strain variation within the composite [66–68]. As thin film-like flexible sensors have relatively small variations in their geometry, the change of SED relative to the external contact force is considered as the main factor affecting the piezoresistive effect. Therefore, in the structure optimization and simulation, the variation of SED is used to evaluate the nominal sensitivity instead of current.

The material properties of the sensors are defined as linear elastic with a Young's Modulus of 3.0 MPa and Poisson's ratio of 0.3. A flat plate is used to compress the sensor to a specified displacement, with the bottom surface of the sensor fixed. A mesh size of 0.5 mm with tetrahedral elements is used for the simulation. "Hard contact" is assigned between the flat plate

and the sensor, and "self-contact" is defined over all the external and internal surfaces of the sensor to measure the self-contact area during compression. The SED is extracted from all the elements to derive the nominal sensor sensitivity. A total of 230 simulations with different auxetic structures are carried out for optimization to find the largest nominal sensitivity under the same external stimuli. The simulation results are also validated against experimental data, with good agreement achieved between the nominal and real sensitivity. Therefore, the FE simulation provides reliable optimization results for the structural design.

**The bone-on-bone load monitoring and sensorimotor control of the biomimetic hand with the tactile sensor**. The sensor is printed onto the lumbar vertebra and the distal femur head, and connected to a customized circuit consisting of two shift registers, two multiplexers, and an Arduino Uno® board for collecting the analog output from the tactile sensing elements. The analog output is then converted into digital signal and visualized through a GUI programmed in Processing IDE (Processing.org).

A tendon-driven biomimetic hand containing intact hand bone skeleton, interphalangeal ligaments, tendon and skin is employed in this study to demonstrate the sensorimotor control with the 3D printed tactile sensor. The skeleton of the hand is 3D printed with polylactic acid (PLA) and the soft-tissues are modeled with silicone-rubber. The anthropomorphic size of the biomimetic hand is reconstructed based on the human hand from a 23-year-old male subject. The sensor is 3D printed onto the distal phalange of the index finger, connected with the same customized electric circuit to collect the analog output from the tactile sensing elements and convert it to digital signals as the feedback for controlling the biomimetic hand. Fishing lines are connected with the bone skeleton and driven by electric motors (Dynamixel MX-12W, Robotics Inc., US) imitating the tendon-driven mechanism of human hand. The pulling force along the fishing line provided by the electric motors is scaled with muscle contraction forces of the human subject through a dynamometer. The in-vivo grasping experiment (see Supplementary, Fig. S14) is carried out and the muscle forces under the reactive touch are derived based on the electromyographic signals collected through the Delsys system (Delsys Trigno, Delsys Inc., US), The experiment details are provided in Supplementary Methods in the supplementary information. The subject gave informed consent to participate in the grasping experiments, which were approved by the Ethics Committee of the First Hospital of Jilin University. A python program (see Supplementary Data 5) is developed for processing the pressure signal and controlling the motors to produce the pulling forces with the preset magnitudes similar with the muscle contraction forces of human subject. The human-like sensorimotor performance is restored on this biomimetic hand based on the tactile feedback, ensuring its application on the next-generation neuro-prosthetic.

## Data availability

The source data, including Supplementary Figs. S1−14, Table 1, 2, and Supplementary Movie 1, are provided with this paper. In Supplementary Movie 1, the 3D printing process of the tactile sensor, the finite element simulations for optimizing the structure, and some sensor measurements are presented. The 3D model and 2D drawings of the proposed tactile sensor can be found on figshare (DOI: 10.6084/m9.figshare.16569696). Data supporting the findings of this manuscript are available from the corresponding author upon reasonable request.

## Code availability

The G code for 3D printing the sensor and the Python code for performing sensorimotor control on biomimetic hand are available on OSF (https://osf.io/p2uyt/?view_only=d62014436e414ce69ab22f0a496033b3).

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

## Acknowledgements
We would like to thank our research group members at the University of Manchester and Jilin University for their great support and assistant to this study.

## Author contributions
Y.W. and L.R. conceived the research idea. Y.W. designed and fabricated the sensor, performed the experiments, analysed the data and wrote the paper. M.D. and Z.Q. provided scientific guidance and revised the paper. B.L., Y.Z. and L.Y. performed some experiments. L.R. and G.W. supervised the research work, analysed the data and revised the manuscript.

## Funding
This research was partly supported by the project of National Key R&D Program of China (No. 2018YFC2001300), the project of National Natural Science Foundation of China (No. 91948302, No. 91848204, No. 52005209, and No. 51675222).

## Competing interests
The authors declare no competing interests.
