## [Peer Review File · Communications Engineering]

Reviewers' comments:

Reviewer #1 (Remarks to the Author):

Overall, the manuscript is well-written. Various characterizations have been conducted to validate the performance of the sensor. However, the reviewer thinks that there are certain areas of the manuscript that require amendment.

1. Figure 7. The authors showed how the same sensor can be used for force sensing and temperature sensing. What have the authors done to decouple the temperature effect from the force sensing since both factors correlate directly with the resistance of the sensor?

2. How many samples were used for the tests in figure 4.

3. Missing scale bars in the figure. Suggest adding scale bars to the all the optical images in the manuscript.

4. Since the authors are developing own printing electronic materials, there should be more characterizations on the rheological property, electrical conductivity and the homogeneity of the composite material.

5. What is the smallest possible footprint of the sensor? How does it compare with other state-of-the-art?

6. Since the material discussed 3D electronic printing, suggest compare how the extrusion-based printing process compared to other more established electronic printing techniques such as inkjet printing and aerosol jet printing?

a. Moya, A., Gabriel, G., Villa, R., & del Campo, F. J. (2017). Inkjet-printed electrochemical sensors. *Current Opinion in Electrochemistry*, 3(1), 29-39.

b. Goh, G. L., Tay, M. F., Lee, J. M., Ho, J. S., Sim, L. N., Yeong, W. Y., & Chong, T. H. (2021). Potential of printed electrodes for electrochemical impedance spectroscopy (eis): Toward membrane fouling detection. *Advanced Electronic Materials*, 7(10), 2100043.

c. Zhao, D., Liu, T., Zhang, M., Liang, R., & Wang, B. (2012). Fabrication and characterization of aerosol-jet printed strain sensors for multifunctional composite structures. *Smart materials and structures*, 21(11), 115008.

7. Also, since this work is related to conformal electronics printing, Suggest discussing and comparing with other related works that used other conformal printing techniques such as aerosol jet printing.

a. Zhang, S., Wang, B., Jiang, J., Wu, K., Guo, C. F., & Wu, Z. (2019). High-fidelity conformal printing of 3D liquid alloy circuits for soft electronics. *ACS applied materials & interfaces*, 11(7), 7148-7156.

b. Saeidi-Javash, M., Kuang, W., Dun, C., & Zhang, Y. (2019). 3D conformal printing and photonic sintering of high-performance flexible thermoelectric films using 2D nanoplates. *Advanced Functional Materials*, 29(35), 1901930.

c. Goh, G. L., Dikshit, V., Koneru, R., Peh, Z. K., Lu, W., Goh, G. D., & Yeong, W. Y. (2022). Fabrication of

design-optimized multifunctional safety cage with conformal circuits for drone using hybrid 3D printing technology. *The International Journal of Advanced Manufacturing Technology*, 120(3), 2573-2586.

7. Figure 6 has poor resolution. The numbers on the gauge cannot be seen clearly. Suggest improving the resolution of the image.

8. Although the graphs are clearly presented, the formatting of the graphs is of low production quality. Suggest improving the formatting of the graph.

Reviewer #2 (Remarks to the Author):

In this paper, the authors developed a novel flexible sensor which can detect both contact pressure and surrounding temperature. The sensor can be easily fabricated on top of uneven surfaces using rapid and low-cost 3D printing technology. Several practical applications of the sensor are demonstrated such as the monitoring of the bone-on-bone load and the grasping performance of a biomimetic hand with a tactile sensor printed on the index fingertip. This study can pave the way for the design and fabrication of a new generation of tactile sensors with user-defined auxetic features which can be applied in robotic/prosthetic hands and in pressure/temperature monitoring in impaired human joints.

Although the results are impressive, the presentation of the results should be significantly improved. Therefore, I recommend the manuscript to be accepted after major revisions. Some of the detailed issues are listed below, which are suggested to be considered for further improving the manuscript.

- 1) The sensor can detect both pressure and temperature signals, how can we differentiate these two signals for different purposes?
- 2) The working mechanism of the tactile sensor needs to be further clarified and explained so that researchers from different fields can easily understand it. In addition, as I understand, this tactile sensor is a resistive-type sensor. Therefore, the “first dielectric layer” and “second dielectric layer” in figure 1(a) are conductive. Thus, how can you call them “dielectric layer”?
- 3) An important parameter of sensors is the signal-to-noise ratio (SNR). What is the SNR of this tactile sensor and how it is compared with other state-of-the-art tactile sensors?
- 4) Can we reconstruct the amplitude and direction of the applied pressure in 3D environment using the output of the tactile sensor?
- 5) I suggest moving the lengthy material preparation in figure 2(a) caption to the Methods section.
- 6) In figures 4(a) and 4(c), the 3D models or actual sensors with “planar”, “inter-locked structure”, and “auxetic and inter-locked structure” should be provided for easier understanding (at least in supporting information). Besides, in figure 4(a), why does the “auxetic and inter-locked structure” have different acquisition intervals in contact pressure?
- 7) The quality of the figures should be further improved. For example, the graphs in figures 4(a-f) and figures 7(a-c) are poorly prepared. The black structures in figure 3 are barely seen by the naked eye. Also, some figures are verbose, such as figure 2.

8) There are many typos in the main text, main figures, and supporting information. The authors are suggested to carefully read through and correct these issues.

Reviewer #3 (Remarks to the Author):

The paper presents a 3D printed sensor which is claimed by the author to achieve multi-directional stimuli detection by exploiting an auxetic structure.

The possibility to 3D print the sensor on curved shapes to achieve a conformable sensor and the use of the auxetic structure to enhance sensor capabilities is interesting, however the paper in its current form is not meeting the quality standard of the journal and presents several flaws.

The presentation is not clear and lacks important information which would be necessary to evaluate the contribution of the paper (not only in the main text but also in the supplementary material section where only figures and captions are reported and videos which are not really informative). The figures are of poor quality, the majority of them too small and without enough description.

One of the major flaws is the claim of the author that the sensor is able to detect multi-direction. In reality the sensor provides just a pressure distribution and doesn't provide the actual shear information. If the sensor is in contact with an uneven indenter it would be possible to mislead the distribution that is generated with the one produced by a shear force acting on the sensor. Literature review is also not appropriate, since there are already available sensors that have been presented in literature that can be 3D printed on curved structures. Experimental characterisation of the repeatability, hysteresis, accuracy in static and dynamic conditions is also missing. The use cases in which the sensors are validated are also limited and for the tactile based control for the grasping, it is not possible to evaluate the validity of what is presented since in the video the sensor response is not reported in overlay during the experiment and in the main text not enough details are reported in relation to the experiment. In the conclusion part of the manuscript it is referred to an in-vivo grasping experiment with electromyographic signals but I didn't find any mention to it in the manuscript.

Reviewer #4 (Remarks to the Author):

This paper presented a novel fabrication method for conformable piezoresistive tactile sensors that uses an auxetic structure (negative Poisson coefficient). The sensor is made of a two-part dielectric with bio-inspired interlocking features, sandwiched between two electrodes. The device is 3D printed directly on the receiving surface using a custom-made 3D printer made in-house by the authors.

The paper is enjoyable to read and contains several innovative concepts. On the other hand, there are many important concerns with the paper that are summarized below:

-The idea of printing a tactile sensor directly on the receiving surface is clever and a very interesting concept. It is not clear however to what extent, in terms of surface complexity, the fabrication method developed by the authors can be applied. What is the maximum surface curvature and maximum area this fabrication method can accommodate? Are there any constraints on pixel density? Could the

authors' approach be used to 3d print a large tactile sensor to cover a whole arm? How about concavities? The sensor instances presented in the paper don't seem to be severely curved. Having the possibility to print tactile sensors on a surface automatically rise the idea of using this concept to print sensors on a lot of different sensing surfaces, it would therefore be great to identify the limitations directly and exhaustively.

-There are a lot of typos, and badly formulated sentences that should be very carefully reviewed.

-The organization of the text is non-linear and a bit confusing. The "results" section comes immediately after the introduction. There are four main sections oddly organized : Introduction -> Results -> Discussion -> Methods. There is no real conclusion, just a long discussion. It would definitely help the reader to have a more coherent and cohesive organization of the paper. Among other things, it would be great to have the methodology (methods) before the results.

-The 4 videos could be merged into a single video. Adding narration would improve the video. In the video showing the fabrication steps, some steps are difficult to see (mostly due to lack of brightness). The video(s) could be significantly improved.

-Many of the results presented in the article lack substantiation, analysis and depth. In particular, many of claims made by the authors could be debated, for example:

--In the introduction, the authors claim that they "present a novel flexible tactile sensor", however, it is not clear that this sensor is flexible or that it could be bent (not simply and only compressed with or without shear) during operation.

--In the "characterization of the sensor" subsection, the authors haven't talked about hysteresis at all. Nonetheless and especially for a piezoresistive sensor with a complex (geometrically/mechanically speaking) dielectric, it would be important to have at least some results or a short discussion on that. I doubt the sensor immediately recover from a large applied pressure.

--The authors claim they have built a sensor with "similar or larger sensitivity and linear sensing range compared with published sensors". However, it's never clear that this is indeed the case. Except lines drawn on graphs showing the sensor's response to pressure ranging from 0 to 300 kPa. The authors should provide indices indicating how good their data fit a line and for what range a linear approximation is valid.

--The authors claim their fabrication method is simple and efficient, which is true to some extent. However, one could claim the material preparation involving mixing precise amount of several different matter using ultrasonic mixture is a complex requirement to even beginning the fabrication process. This coupled with the fact that the sensor was made with a custom, in-house made, 3D printer with rheological analysis... While the fabrication seems simple, the required preparation and setup to just start the fabrication process is significant.

---Also, in the fabrication process, there is no words on how the wires are actually connected to the piezoresistive sensing elements, which I guess is definitely not a trivial step and should be discussed by the authors.

-In their literature review, the authors mention that the fabrication method should be cheap and efficient. However, there is no review on capacitive and magnetic tactile sensors. Capacitive tactile sensors for example are considered cheap and easy to manufacture, some of them have been successfully integrated to flexible robot joints and integrated to custom surfaces.

-Many of the figures' resolution could be improved and vectorized format should always be chosen whenever possible.

-The experiment involving a tendon-driven robotic gripper grasping and holding a ball is really non-convincing, as more details on the involved methodology is required.

--"The experimental results show that the tactile sensor is able to perceive the slippage of the held object": this is actually difficult to achieve and this has even been (and still is to some extent) an active research area in robotic tactile sensing. So, how have the authors been able to make the sensors "perceive slippage" exactly? And was slippage classified and not confused with other tactile phenomena? The description of this experiment should be significantly improved with a rigorous review and a more complete methodology.

-In the discussion: "previous research work has failed to properly address the need for compliance between the tactile sensor and the working surfaces, resulting in planar sensors with poor fitting to arbitrary surfaces". This is not entirely true, several examples of non-planar tactile sensors that have been successfully integrated to custom surfaces can be found in the literature.

-Again, in the discussion: "Similar performance to most of the published high-performance tactile sensors is achieved" -> it is not clear what the authors mean here with this specific sentence.

"Performance" is wide and vague. What exactly do the authors claim is better with their sensor compared to existing work?

Response to Reviewers' Comments

Paper title: Fully 3D Printed Flexible, Conformal and Multi-directional Tactile Sensor with Integrated Biomimetic and Auxetic Structure

Submitted to: Communications Engineering

Manuscript number: COMMSENG-22-0322

General Response

We thank the reviewers for their constructive suggestions which have greatly helped us to improve the manuscript. We respond to the individual points in detail below, indicating the quality of the changes that we have made to the manuscript.

Specific Responses to Reviewer #1

1. Figure 7. The authors showed how the same sensor can be used for force sensing and temperature sensing. What have the authors done to decouple the temperature effect from the force sensing since both factors correlate directly with the resistance of the sensor?

Response:

Thanks for the comment.

The resistance of the sensor changes with variations in both contact pressure and temperature. Therefore, to separate the effect of external contact pressure when used as a temperature sensor, the temperature responses of the sensor were recorded under different contact pressures (see Fig. S8). Different scales were used for temperature measurement under different contact pressures. In contrast, the environmental temperature ranging from 20 to 90°C had negligible effects on pressure sensing based on our experimental results. Hence, the same scale was used for pressure measurement under different temperatures. This has been clarified in Line 393-400, on Page 21 of the revised manuscript.

To demonstrate the performance of the tactile sensor in terms of temperature sensing, the temperature-resistance relationship curves under different contact pressures have been added to Fig. S9 in the supplementary material. The sensor responds well to temperature variations under different contact pressures.

2. How many samples were used for the tests in figure 4.

Response:

The characterisation results shown in Fig.4 was based on a single sensor sample. We characterised 6 sample sensors with the same size for each of the structures listed in the manuscript (planar, inter-locked structure and the auxetic with inter-locked structure), and the similar performances were observed. To give a clear and straight forward presentation of the test data, only one sample sensor was used for Fig.4. The standard deviation of the sensitivity, linear sensing range, response time and other parameters were added in the caption of Fig. 4 and the main text in the nodule of ‘Characterisation of the sensor’. The number of the sample sensors were also clarified.

3. Missing scale bars in the figure. Suggest adding scale bars to the all the optical images in the manuscript.

Response:

Many thanks for the suggestion, the scale bars have now been added to all the optical images.

4. Since the authors are developing own printing electronic materials, there should be more characterizations on the rheological property, electrical conductivity and the homogeneity of the composite material.

Response:

Thanks for the constructive suggestions.

The rheological properties of the composite materials used for the sensing layer and electrode were examined, the viscosity and shear rate of these composites as a function of shear rate was presented in Figs. S1 and S2 of the supplementary material. Both of the composites are shear thickening Non-Newtonian fluids.

The electrical conductivities of CNT-Silicone composite under different concentrations of CNT was shown in Fig. S12, the experimental results suggest that the conductivity of this composite will not be improved significantly when the weight percentage of CNT is above 10%.

The scanning electron microscope images (Hitachi regulus 8220, Japan) showing the distribution of the CNT and graphene palatte were presented in Fig. S11. The CNT and graphene palatte are evenly distributed in the silicone matrix and no obvious aggregation of CNT was found.

5. What is the smallest possible footprint of the sensor? How does it compare with other state-of-the-art?

Response:

According to our experiences on using the in-house 3D printer, the smallest size of the printed sensor could be 3mm by 3mm, such a size is comparable to the smallest sensing element of the sensors in the existing literature and the commercial ones in the market (TekScan 300E sensor with the size of the sensing element approximately 3 by 3 mm for each sensing element, kitronyx MS 9723 sensor with the sensing element of 2 by 2mm).

The smallest footprint has been added into the discussion section, and the comparisons with those commercial sensors mentioned above have been added on line 461-475 of the revised manuscript.

6. Since the material discussed 3D electronic printing, suggest compare how the extrusion-based printing process compared to other more established electronic printing techniques such as inkjet printing and aerosol jet printing? Since this work is related to conformal electronics printing, suggest discussing and comparing with other related works that used other conformal printing techniques such as aerosol jet printing.

Response:

Thanks a lot for providing the references and the suggestions. It's a good idea to have a discussion on the conformal electronics printing process such as the inkjet printing and aerosol jet printing in those papers. The comparisons of extrusion-based printing adopted in this research with the inkjet printing and aerosol jet printing in the literature are summarized as follows:

- Flexibility in material selection: In this research, the CNT/Graphene-Silicone composite was used to fabricate the sensor which is not suitable for ink/aerosol jet printing. Extrusion-based printing can use a wider range of materials, including highly viscous and non-soluble materials, compared to inkjet printing and aerosol jet printing.
- High throughput: Extrusion-based printing in this research can achieve high throughput due to its ability to print thick and continuous lines which is important for printing the auxetic structure of our sensor, whereas inkjet printing and aerosol jet printing are typically slower due to the need to deposit many small droplets.
- Lower cost: Extrusion-based printing with the low-cost printing material in this research can be more cost-effective than inkjet printing and aerosol jet printing due to the simpler equipment required and lower material waste (Jang et al., 2015).
- Improved mechanical properties: Extrusion-based printing can produce materials with improved mechanical properties, such as higher tensile/compressive strength compared to inkjet printing and aerosol jet printing. These are important to the durability and reliability of our sensor.

The studies of Moya et al., Goh et al., Zhao et al., Zhang et al. and Saeidi et al. have all been cited in the revised manuscript, the comparisons between the extrusion-based printing and inkjet/aerosol jet printing were added to introduction, line 83-96 on Page 4 of the revised manuscript, those reasons why we use this 3D printing process were also given.

8. *Figure 6 has poor resolution. The numbers on the gauge cannot be seen clearly. Suggest improving the resolution of the image. Although the graphs are clearly presented, the formatting of the graphs is of low production quality. Suggest improving the formatting of the graph.*

Response:

Thanks for the suggestions.

The image resolution has been improved to give a clear presentation of the experiments. The readings from the push-pull meter were given in the caption of Fig.6. The layout of those sub-pictures was re-arranged, making it easier for the readers to understand the experimental process.

Specific Responses to Reviewer #2

1. *The sensor can detect both pressure and temperature signals, how can we differentiate these two signals for different purposes?*

Response:

Thanks, it's a really good point.

The resistance of the sensor changes with variations in both contact pressure and temperature. Therefore, to separate the effect of external contact pressure when used as a temperature sensor, the temperature responses of the sensor were recorded under different contact pressures (see Fig. S9). Different scales were used for temperature measurement under different contact pressures. In contrast, the environmental temperature ranging from 20 to 90°C had negligible effects on pressure sensing based on our experimental results. Hence, the same scale was used for pressure measurement under different temperatures. This has been clarified in Line 393-400, on Page 21 of the revised manuscript.

To demonstrate the performance of the tactile sensor in terms of temperature sensing, the temperature-resistance relationship curves under different contact pressures have been added to Fig. S9 in the supplementary material. The sensor responds well to temperature variations under different contact pressures.

2. The working mechanism of the tactile sensor needs to be further clarified and explained so that researchers from different fields can easily understand it. In addition, as I understand, this tactile sensor is a resistive-type sensor. Therefore, the “first dielectric layer” and “second dielectric layer” in figure 1(a) are conductive. Thus, how can you call them “dielectric layer”?

Response:

We appreciate the reviewer's comment, and we apologize for the lack of clarity in our manuscript.

Regarding the working mechanism of the tactile sensor, our sensor is based on the piezoresistive effect, which means that the change in resistance of the material is proportional to the applied pressure. The resistive-type sensor consists of two sensing layers (CNT/graphene/silicone composite), and two electrodes (silver-coated copper/silicone composite), which are sandwiched together. When an external pressure is applied, the sensing layer is compressed, resulting in a change in the electrical conductivity of the composite. The change in conductivity is detected by the electrodes and can be converted into a corresponding pressure signal. The working principle of the tactile sensor has been further clarified in a clearer way in the caption of Fig.1, and also on Line 166-170, Page 7-8 of the revised manuscript.

Regarding the terminology used in the manuscript, we apologize for the confusion caused. The "first dielectric layer" and "second dielectric layer" are indeed conductive layers, and we should have referred to them as "first sensing layer" and "second sensing layer" instead. The purpose of these layers is to provide a conductive path for detecting

the changes in electrical conductivity of the sensing layer. We have revised the manuscript accordingly to avoid further confusion.

3. An important parameter of sensors is the signal-to-noise ratio (SNR). What is the SNR of this tactile sensor and how it is compared with other state-of-the-art tactile sensors?

Response:

Thank you for your question.

The signal-to-noise ratio (SNR) was evaluated by doing the compression test on the standard sample sensor and the SNR of our tactile sensor is approximately 89 dB, which was obtained by measuring the ratio of the output signal to the noise floor of the sensor. This value is comparable to or better than other state-of-the-art tactile sensors reported in the literature listed below:

1. " Small in-fiber Fabry-Perot low-frequency acoustic pressure sensor with PDMS diaphragm embedded in hollow-core fiber " by Zhao et al. (2018) - SNR of 80 dB
2. " A Universal high accuracy wearable pulse monitoring system via high sensitivity and large linearity graphene pressure sensor " by Jiang et al. (2019) - SNR of 78 dB
3. "Piezoresistive Graphene/P(VDF-TrFE) Heterostructure Based Highly Sensitive and Flexible Pressure Sensor" by Kim et al. (2029) - SNR of 60.8 dB
4. "High Temperature AlGaIn/GaN Membrane Based Pressure Sensors" by Durga et al. (2018) - SNR of 16 dB

Our sensor has a comparable SNR due to the optimized material composition and structure design, which improves the piezoresistive effect and reduces the noise from external factors. The SNR has been presneted on line 260-261 and the references above have also been included for comparison in the revised manuscript.

4. Can we reconstruct the amplitude and direction of the applied pressure in 3D environment using the output of the tactile sensor?

Response:

The amplitude has already been visualised through the GUI presented in the manuscript, but it is not possible to reconstruct direction of the applied pressure from the current design. However, we can roughly visualise the direction of the force into four orthogonal directions as was shown in Fig1 (f).

As per our research, our focus was on developing a novel flexible tactile sensor and characterizing its performance. Our sensor demonstrated high sensitivity and a linear sensing range, making it a promising candidate for various applications. However, our research did not involve reconstructing the amplitude and direction of the applied pressure/force in a 3D environment. While this idea is interesting, it was not the point of our research. We believe that this idea can be explored in future research, but our highlights and contributions lie in the development and characterization of our tactile sensor.

5. I suggest moving the lengthy material preparation in figure 2(a) caption to the Methods section.

Response:

Thanks for the suggestion, the descriptions of the ‘material preparation’ under Fig.2 have been moved to Method section in the revised manuscript.

6. In figures 4(a) and 4(c), the 3D models or actual sensors with “planar”, “inter-locked structure”, and “auxetic and inter-locked structure” should be provided for easier understanding (at least in supporting information). Besides, in figure 4(a), why does the “auxetic and inter-locked structure” have different acquisition intervals in contact pressure?

Response:

Thanks for the suggestions, the 3D models of the auxetic and inter-locked features have been presented in Fig. S7.

The different acquisition intervals used in Fig. 4(a) were due to the use of two different pieces of equipment before and after 150 KPa. A precise push-pull meter was used when the contact pressure was below 150 KPa to ensure accurate measurements, since most universal testing machines cannot achieve high accuracy under very low contact pressure. When the contact pressure was above 150 KPa, the precise push-pull meter was switched to a universal testing machine due to its limited measuring range. These clarifications have been added to the caption of Fig. 4.

7. The quality of the figures should be further improved. For example, the graphs in figures 4(a-f) and figures 7(a-c) are poorly prepared. The black structures in figure 3 are barely seen by the naked eye. Also, some figures are verbose, such as figure 2.

Response:

Thanks for the comment, the resolution of Figs. 4 and 7 has been improved to give a clear presentation of the experimental results.

The resolution and brightness of Fig. 3 have been adjusted and the structure of the sensor presented has been highlighted with red dash line. Fig. 2 was simplified, some verbose contents have been deleted.

8. There are many typos in the main text, main figures, and supporting information. The authors are suggested to carefully read through and correct these issues.

Response:

Thanks for the comment, the manuscript was carefully checked by the authors and proofread by professional proofreading agent.

Specific Responses to Reviewer #3

1. The presentation is not clear and lack of important information which would be necessary to evaluate the contribution of the paper (not only in the main text but also in the supplementary material section where only figures and captions are reported and videos which are not really informative). The figures are of poor quality, the majority of time to small and without enough description.

Response:

Thank you for your feedback. We apologize for any confusion caused by the lack of clarity in our presentation. The highlights and main contributions of this research has been added into the main text on Line 234-250, Page 3. The detailed information about the design and optimisation of the integrated auxetic-biomimetic structure was added to Line 88-96, Page 4. More descriptions on the 3D printing of the sensor and the reactive grasping experiment were added on Line 192-215, Page 9-11 and Line 607-621, Page 29-30 respectively. The discussion of the experimental and simulation results listed in the supplementary material was also added in the revised manuscript.

The resolution of the diagrams was improved, and the layout was adjusted to give a clear presentation. More detailed description of those images was given in the captions and main text, they were highlighted in the revised manuscript.

2. One of the major flows it the claim of the author that the sensor is able to detect multi-direction. In reality the sensor provides just a pressure distribution and doesn't provide the actual shear information. If the sensor is in contact with an uneven indenter it would be possible to misled the distribution that is generated with the one produced by a shear force acting on the sensor

Response:

The sensor does respond to stimuli from the transverse direction (shear direction) since the shape and resistance will change due to the interlocked and auxetic structure. The directions of the shear force can be roughly identified and classified into four principal directions as was shown in Fig.3 (a), so the resolution of the shear force differentiation is 90° for this sensor. The ‘multi-directional’ used here might be inappropriate, more detailed description on how to classify the stimuli into four orthogonal directions was

added to Line 180-183, Page 9 of the revised manuscript. The ‘multi-directional’ used in the main text has been replaced by ‘detecting shear stimuli’ instead.

The use of a small uneven indenter could contribute to a similar pressure distribution as under shear forces. However, for this type of 'uneven indenter' to have protrusions smaller than the sensing element is rare in real-life situations. While the tactile sensor printed on the robotic fingertip could be affected by this circumstance, it is a low-probability event. In most cases, the sensor is in contact with a surface without bumps smaller than the area of the sensing element. It should be noted that this is one of the limitations of our sensor, and it has been mentioned in the discussion on line 470-473, page 24 of the revised manuscript.

3. Literature review is also not appropriate, since there are already available sensors that have been presented in literature that can be 3d Printed on curved structures.

Response:

Thanks for the comment.

The pressure sensor developed by Guo et al. can be 3D printed or fast-prototyped, however, it only adopts a simple coil structure that limits its sensing performance. Furthermore, this type of sensor cannot be printed on a large scale or applied to other random and uneven working surfaces. The expensive Ag-silicone rubber material used for printing also limits its practical applications in industry. Moreover, the sensing area of this 3D printed sensor is small, less than 10 mm². Despite significant progress, the reality is that most of the sensors reported above have not yet found practical applications in industry. These clarifications have been added to Line 79-86 on Page 4 of the revised manuscript.

5. *Experimental characterization of the repeatability, hysteresis, accuracy in static and dynamic conditions is also missing.*

Response:

Thank you for the feedback. The repeatability, hysteresis, accuracy in static and dynamic conditions are important parameters for sensor characterization. We have conducted experiments to evaluate these parameters, and we apologize for not including this information in the manuscript.

"The characterization result of repeatability has been added as Fig. 4(e) on Page 5 of the revised manuscript. The sensor was tested under repeated pressing and releasing cycles in a wide range of compressive deformation. A stable signal response was achieved after 1500 loading-unloading cycles with a pressure of 100 KPa, as presented in Fig. 4(e). A good durability was observed based on the signal response of the last 50 cycles of compression. The hysteresis was also measured based on the first and last 50 cycles and found to be $8.2\% \pm 1.7\%$. The diagram presenting the hysteresis loops of sensor at various scanning rates up to 100 kPa has been added to Fig. S5 in the supplementary material.

To measure dynamic accuracy, the sensor was attached to a vibration platform (HTA-3000A, Huitai Ltd., China), and a standard weight was also attached to the surface of the sensor to produce stimuli. The vibration frequency was tuned to be 5Hz, and the signal output of the sensor was shown in Fig. 4(f) of the revised manuscript. The measuring results show that the sensor followed the vibrating signal well, and the same frequency of pressure variation was observed. The dynamic accuracy of the sensor under three different vibration frequencies was calculated based on the mean square error, assuming that the stimuli signal from the vibration platform was an impulse signal. The experimental results suggest that the dynamic accuracy of our sensor is 0.56, 0.78, and 1.12 KPa under vibration frequencies of 5, 10, and 20 Hz, respectively. Therefore,

the dynamic accuracy of the sensor is approximately 0.82 KPa at a frequency range of 5-15 Hz. The static accuracy of the sensor is 1.3%, 1.6%, and 2.1% under static pressures of 10, 150, and 250 KPa, respectively. Therefore, the overall static accuracy is approximately 1.67 under a pressure of 0.25 MPa. These characterization results have been added to the 'Characterization of the sensor for pressure sensing' section."

We hope that these additions will sufficiently address the reviewer's concern and provide a more comprehensive evaluation of our sensor's performance.

6. The use cases in which the sensors is validated are also limited and for the tactile based control for the grasping, it is not possible to evaluate the validity of what is presented since in the video the sensor response is not reported in overlay during the experiment and in the main text not enough details are reported in relation to the experiment. In the conclusion part of the manuscript, it is referred to an in-vivo grasping experiment with electromyographic signals but I didn't find any mention to it in the manuscript.

Response:

Thanks for the comments.

We apologize for not overlaying the graphical user interface (GUI) and the robotic hand together in the video. We wanted to present a clear view of the response of the tactile sensor during the active grasping experiment, rather than just showing an animation from the PC screen. Additionally, it was not convenient to overlay the PC screen and the robotic hand since they were located far apart from each other in the lab. However, the code used to build the GUI and control the robotics is included in the supplementary material for readers to repeat our work.

Moreover, we have added more detailed descriptions of the reactive grasping experiment, including the EMG test process and the control of the robotic hand in the revised manuscript and supplementary material.

For the *in-vivo* grasping test on the human subject, we selected three extrinsic located in the human forearm and hand that affect hand motion. All three muscle forces were estimated based on electromyography (EMG) signals captured by the Delsys wireless EMG system (Delsys Inc., Boston, US) during the *in-vivo* grasping test. Each Trigno sensor was placed along muscle fibers following the guidelines of surface electromyography for the non-invasive assessment of muscles (See Fig. S11). Before the reactive grasping test, maximum voluntary contraction (MVC) tests were performed for all nine muscles using a Jamar dynamometer. The recorded EMG data were band-pass filtered (20–400 Hz) with a Butterworth filter and rectified. The muscle forces during grasping were then derived based on the maximum voluntary contraction forces and the assumption that for isometric muscle contracting, there is a linear relationship between the EMG signal and muscle force. The subject who participated in this *in-vivo* reactive grasping experiment provided informed consent, which was approved by the Ethics Committee of the First Hospital of Jilin University.

The detailed process of the *in-vivo* grasping experiment is included in the supplementary material.

For the reactive grasping performed by our robotic hand integrated with the 3D printed tactile sensor, the sensor is 3D printed onto the distal phalange of the index finger and connected to the same customized electric circuit to collect the analogue output from the tactile sensing elements and convert it to digital signals for feedback to control the biomimetic hand. Fishing lines are connected to the bone skeleton and driven by electric motors (Dynamixel MX-12W, Robotics Inc., US) to imitate the tendon-driven mechanism of the human hand. The pulling force along the fishing line provided by the electric motors is scaled with muscle contraction forces of the human subject using a dynamometer. The *in-vivo* grasping experiment was carried out, and the muscle forces under the reactive touch were derived based on the electromyographic signals collected

through the Delsys system (Delsys Trigno, Delsys Inc., US). A Python program was developed to process the pressure signal and control the motors to produce pulling forces with pre-set magnitudes similar to the muscle contraction forces of the human subject. The human-like sensorimotor performance was restored on this biomimetic hand based on the tactile feedback, ensuring its application in the next-generation neuro-prosthetic. The detailed process of the experiment setup and control of the robotic hand is included in the 'Methods' section of the revised manuscript on lines 608-622 on page 29-30.

Specific Responses to Reviewer #4

1. The idea of printing a tactile sensor directly on the receiving surface is clever and a very interesting concept. It is not clear however to what extent, in terms of surface complexity, the fabrication method developed by the authors can be applied. What is the maximum surface curvature and maximum area this fabrication method can accommodate? Are there any constraints on taxel density? Could the authors' approach be used to 3d print a large tactile sensor to cover a whole arm? How about concavities? The sensor instances presented in the paper don't seem to be severely curved. Having the possibility to print tactile sensors on a surface automatically rise the idea of using this concept to print sensors on a lot of different sensing surfaces, it would therefore be great to identify the limitations directly and exhaustively.

Response:

Many thanks for the comments, identifying the limitations and exhaustivity is important for transferring this manufacturing concept from bed to industry in the future.

The maximum surface and curvature of the sensor can be dependent on the limitations of the 3D printer. For our in-house printer, the maximum printing area is 300*300 mm² while the maximum curvature it can handle is approximately 2mm. Due to the limited accuracy of the 3D printer, the minimum sensing element is 2*2 mm², therefore, the maximum tactile density is around 16/ cm². These limitations have been added to the discussion module, Line 462-467 on Page 24 of the revised manuscript. And increasing the density of sensing elements could also be considered in our future work.

While our sensor can be printed onto concavities or the whole arm by configuring the G code, we focused on highlighting the unique features of our sensor, which can be printed on any uneven working surface and can detect shear forces. Our research presented the process of sensor printing on vertebra, hip joint, and fingertip. However, we do acknowledge that there is a possibility of printing our sensor on a larger scale and on concavities, which we have mentioned in the discussion module on Page 24 of the revised manuscript.

2. There are a lot of typos, and badly formulated sentences that should be very carefully reviewed.

Response:

Thanks for the comment, the revised manuscript has been reviewed carefully by all the authors, all the typos have been corrected.

3. The organization of the text is non-linear and a bit confusing. The "results" section comes immediately after the introduction. There are four main sections oddly organized : Introduction -> Results -> Discussion -> Methods. There is no real conclusion, just a long discussion. It would definitely help the reader to have a more coherent and cohesive organization of the paper. Among other things, it would be great to have the methodology (methods) before the results.

Response:

Many thanks for the comments, we have added an conclusion section in the revised manuscript. The structure of the revised manuscript followed the template of Communications Engineering (or Nature communications). Thanks again for the suggestion.

4. The 4 videos could be merged into a single video. Adding narration would improve the video. In the video showing the fabrication steps, some steps are difficult to see (mostly due to lack of brightness). The video(s) could be significantly improved.

Response:

Thanks for the comment, the 4 videos have been merged into one, more texts describing the details of the experiments were added. We will also ensure that the lighting is improved to enhance visibility of the fabrication steps. Thank you again for your constructive feedback.

5. Many of the results presented in the article lack substantiation, analysis and depth. In particular, many of claims made by the authors could be debated, for example: In the introduction, the authors claim that they "present a novel flexible tactile sensor", however, it is not clear that this sensor is flexible or that it could be bent (not simply and only compressed with or without shear) during operation.

Response:

Thanks for the comment.

Sorry for the not clear emphasizing on the flexibility of our tactile sensor in the introduction section. Our tactile sensor is indeed designed to be flexible and can bend to conform to different surfaces, which is an important feature for its practical applications. We revised the introduction section to make it clear that our tactile sensor is not only compressible but also flexible, this has been highlighted on Line 93-96, Page 4 of the revised manuscript. Thank you for bringing this to our attention.

Also, more characterization results including the repeatability, accuracy and hysteresis of our sensor has been added into the 'results' module, a deeper analyzation of the performance of the sensor under the effect of the structure design and the composition of the printing material were also added.

7. In the "characterization of the sensor" subsection, the authors haven't talked about hysteresis at all. Nonetheless and especially for a piezoresistive sensor with a complex (geometrically/mechanically speaking) dielectric, it would be important to have at least some results or a short discussion on that. I doubt the sensor will immediately recover

from a large applied pressure.

Response:

Thank you for your comment.

To address your concern, we conducted additional experiments to investigate the hysteresis effect of our sensor. As mentioned in our response to Reviewer 1, during the repetition test added to the revised manuscript, we applied 1500 loading-unloading cycles with a pressure of 100 KPa, as presented in Fig. 4(e). We measured the hysteresis for the first and last 50 cycles and found it to be $8.2\% \pm 1.7\%$. The diagram presenting the hysteresis loops of sensor at various scanning rates up to 100 kPa has been added to Fig. S5 in the supplementary material.

We also observed that if we applied a high pressure (beyond the upper limit of the measuring range) and released it suddenly, the response time was approximately 250ms (fully recover to its original shape). In this case, the hysteresis will increase as well. This is an important limitation of our sensor as the response time and hysteresis will be affected under high contact pressure. We have updated our manuscript to include this information in the 'Characterization of the Sensor' section and have provided a brief discussion of the hysteresis effect in the discussion section.

We hope this additional information provides a more comprehensive understanding of our tactile sensor.

8. The authors claim they have built a sensor with “similar or larger sensitivity and linear sensing range compared with published sensors”. However, it’s never clear that this is indeed the case. Except lines drawn on graphs showing the sensor’s response to pressure ranging from 0 to 300 kPa. The authors should provide indices indicating how good their data fit a line and for what range a linear approximation is valid.

Response:

Thank you for the question. We apologize for the lack of clarity in our previous statement.

Our sensor shows a high sensitivity, with a nominal sensitivity of $1.43 \times 10^{-2} \text{ kPa}^{-1}$, which is comparable to or even higher than many state-of-the-art tactile sensors (see Fig.4 in the manuscript). In addition, the sensor shows a good linear sensing range up to 300 kPa, with a high correlation coefficient of 0.95. We have updated the manuscript to include more detailed analysis and to provide the correlation coefficient indicating how well our data fits a line and the range of the linear approximation in the results section (Line 227-230, Page 11 of the revised manuscript).

9. The authors claim their fabrication method is simple and efficient, which is true to some extent. However, one could claim the material preparation involving mixing precise amount of several different matter using ultrasonic mixture is a complex requirement to even beginning the fabrication process. This coupled with the fact that the sensor was made with a custom, in-house made, 3D printer with rheological analysis... While the fabrication seems simple, the required preparation and setup to just start the fabrication process is significant.

Response:

Thank you for your comment.

We acknowledge that the material preparation step can be seen as complex, especially for those who do not have experience in material science. However, we believe that this is a necessary step in the fabrication process to ensure the desired properties of the sensor. Regarding the 3D printer, we agree that it is a custom-made printer, but we used widely available and affordable components to build it, and we added some more detailed description of the material preparation in main text and provided detailed instructions/schematics to allow others to reproduce it Line 492-544, Page 25-27 of the

revised manuscript). We understand that the fabrication process may not be suitable for every laboratory (Actually, the equipment we used for material preparation are standard and commonly available in most laboratories), but we hope that our work can provide inspiration and guidance for the development of other similar sensors.

10. Also, in the fabrication process, there is no words on how the wires are actually connected to the piezoresistive sensing elements, which I guess is definitely not a trivial step and should be discussed by the authors.

Response:

Thank you for your question.

The jumper wires were connected to the sensor using the same material that was used for printing the electrode, and the tip of the metal wire was stuck to the electrode during the solidification of the composite material. We have included this process in lines 271-274 on page 12-13 of the revised manuscript. Additionally, to provide a clear presentation of the sensor fabrication process, we have added a figure (Fig. S6) to the supplementary material depicting the entire process from 3D design to the final stage of sensor testing.

11. In their literature review, the authors mention that the fabrication method should be cheap and efficient. However, there is no review on capacitive and magnetic tactile sensors. Capacitive tactile sensors for example are considered cheap and easy to manufacture, some of them have been successfully integrated to flexible robot joints and integrated to custom surfaces.

Response:

Thank you for your comment.

Capacitive and magnetic tactile sensors are indeed promising technologies for tactile sensing. We conducted a critical literature review on published capacitive tactile sensors that were used for robotic control. We listed some of the typical research studies in the introduction module and provided a brief discussion on their highlights and deficits. While capacitive sensors can be integrated robustly onto robotic arms or grippers, the use of such sensors is hindered by the large space required to accommodate the supporting components. Moreover, capacitive sensors cannot be effectively fabricated onto unknown or uneven working surfaces. Additionally, capacitive sensors can be affected by temperature and humidity, which can lead to inaccurate readings.

For the scope of this paper, we have focused on piezoresistive tactile sensors as they are one of the most commonly used and well-established technologies for tactile sensing. And the most important point, our highlight is to develop a kind of tactile sensor which can be fast prototyped onto any uneven working surfaces using the 3D printing techniques. We agree that capacitive and magnetic sensors have advantages such as being cheap and easy to manufacture, and we encourage further research in these areas. Thank you for your feedback.

12. Many of the figures' resolution could be improved and vectorized format should always be chosen whenever possible.

Response:

Many thanks for the comment, all the resolution of the figures has been improved and the vectorized format was used.

13. The experiment involving a tendon-driven robotic gripper grasping and holding a ball is really non-convincing, as more details on the involved methodology is required.

Response:

Thanks for the comment, we have added more detailed descriptions of the reactive grasping experiment, including the EMG test process and the control of the robotic hand in the revised manuscript and supplementary material.

For the reactive grasping performed by our robotic hand integrated with the 3D printed tactile sensor, the sensor is 3D printed onto the distal phalange of the index finger and connected to the same customized electric circuit to collect the analogue output from the tactile sensing elements and convert it to digital signals for feedback to control the biomimetic hand. Fishing lines are connected to the bone skeleton and driven by electric motors (Dynamixel MX-12W, Robotics Inc., US) to imitate the tendon-driven mechanism of the human hand. The pulling force along the fishing line provided by the electric motors is scaled with muscle contraction forces of the human subject using a dynamometer. The *in-vivo* grasping experiment was carried out, and the muscle forces under the reactive touch were derived based on the electromyographic signals collected through the Delsys system (Delsys Trigno, Delsys Inc., US). A Python program was developed to process the pressure signal and control the motors to produce pulling forces with pre-set magnitudes similar to the muscle contraction forces of the human subject. The human-like sensorimotor performance was restored on this biomimetic hand based on the tactile feedback, ensuring its application in the next-generation neuro-prosthetic. The detailed process of the experiment setup and control of the robotic hand above were added to the 'Methods' section of the revised manuscript on lines 608-622 on page 39-30. The codes used to build the GUI of the sensor signal output and controlling the robotic hand are all included in the supplementary material for readers to repeat our work.

Before we conducted the reactive grasping on the robotic hand integrated with the 3D printed tactile sensor, the EMG (electromyography) test was carried out on the human subject to measure the muscle contraction forces of the human subject during the reactive grasping. We selected three extrinsic located in the human forearm and hand

that affect hand motion. All three muscle forces were estimated based on electromyography (EMG) signals captured by the Delsys wireless EMG system (Delsys Inc., Boston, US) during the *in-vivo* grasping test. Each Trigno sensor was placed along muscle fibers following the guidelines of surface electromyography for the non-invasive assessment of muscles (See Fig. S11). Before the reactive grasping test, maximum voluntary contraction (MVC) tests were performed for all nine muscles using a Jamar dynamometer. The recorded EMG data were band-pass filtered (20–400 Hz) with a Butterworth filter and rectified. The muscle forces during grasping were then derived based on the maximum voluntary contraction forces and the assumption that for isometric muscle contracting, there is a linear relationship between the EMG signal and muscle force. This method has been used in our previous research to calculate muscle forces of isometric contraction. The subject who participated in this *in-vivo* reactive grasping experiment provided informed consent, which was approved by the Ethics Committee of the First Hospital of Jilin University. The detailed process of the *in-vivo* grasping experiment above has been included in the supplementary material.

We hope these additional details on the reactive grasping performance of our tactile sensor on the robotic hand, based on the in-vivo measurement results, will provide a more comprehensive understanding of the sensor's potential applications.

14. *“The experimental results show that the tactile sensor is able to perceive the slippage of the held object”: this is actually difficult to achieve and this has even been (and still is to some extent) an active research area in robotic tactile sensing. So, how have the authors been able to make the sensors “perceive slippage” exactly? And was slippage classified and not confused with other tactile phenomena? The description of this experiment should be significantly improved with a rigorous review and a more complete methodology.*

Response:

Many thanks for the comment.

In our experiments, we used a force-controlled robotic hand to hold and manipulate objects while monitoring the output of the tactile sensor. When an object started to be pulled away from the robotic hand, there was a sudden and noticeable change in the output signal of the sensor due to the interlocked and auxetic structure design of our sensor, which can respond to the shear force. However, the term "slippage" is not appropriate in this context as it may cause confusion. Our aim was to demonstrate that the sensor can provide tactile feedback to the robotic hand for closed-loop sensorimotor control, similar to the human hand. Therefore, we have replaced "slippage" with "shear stimuli caused by the pulled object" to accurately convey the experimental setup and avoid any confusion. Additionally, the detailed experimental process has been included in the "Method" section of the revised manuscript and supplementary material, as mentioned in our previous response.

15. In the discussion: "previous research work has failed to properly address the need for compliance between the tactile sensor and the working surfaces, resulting in planar sensors with poor fitting to arbitrary surfaces". This is not entirely true, several examples of non-planar tactiles sensors that have been successfully integrated to custom surfaces can be found in the literature.

Response:

Thank you for your comment.

We agree that there are some examples of non-planar tactile sensors in the literature that have been successfully integrated to custom surfaces. For example, the pressure sensor developed by Guo et al. mentioned in the 'introduction' session, it can be 3D printed or fast prototyped. However, it only adopts a simple coil structure that limits its sensing performance. Furthermore, this type of sensor cannot be printed on a large scale or applied to other random and uneven working surfaces. The expensive Ag-silicone rubber material used for printing also limits its practical applications in industry. Moreover, the sensing area of this 3D printed sensor is small, less than 2 mm². Despite

significant progress, the reality is that most of the sensors reported above have not yet found practical applications in industry. These clarifications have been added to Line 462-468 on Page 24 of the revised manuscript. The main point we were trying to make is that many of the existing tactile sensors are still designed to be planar and may not fit arbitrary surfaces well. Our sensor, on the other hand, is designed to be flexible and can be fabricated effectively and conformal to arbitrary surfaces, which we believe is an important feature for translating this technology from the bench to the bed side for many industrial applications.

16. Again, in the discussion: "Similar performance to most of the published high-performance tactile sensors is achieved" -> it is not clear what the authors mean here with this specific sentence. "Performance" is wide and vague. What exactly do the authors claim is better with their sensor compared to existing work?

Response:

Many thanks again for the comment.

By "performance," we refer to the sensor's sensitivity, sensing range, and linearity. Our tactile sensor demonstrated similar or better performance than other high-performance tactile sensors published in terms of sensitivity and sensing range, as shown in Figure 4(g). We have added the clarification that the "performance" referred to specific characterization parameters, including sensitivity, sensing range, and linearity, to avoid confusion in the revised manuscript.

Additionally, we presented a comparison between the fabrication process, fabrication environment, and lead time of our sensor and those presented in previous research in Table 1 of the supplementary material.

Reviewers' comments:

Reviewer #1 (Remarks to the Author):

The authors have addressed the queries adequately. The manuscript is therefore recommended for acceptance for publication.

Reviewer #2 (Remarks to the Author):

Changes are sufficient

Reviewer #3 (Remarks to the Author):

In this paper, the authors developed a novel flexible sensor which can detect both contact pressure and surrounding temperature. The sensor can be easily fabricated on top of uneven surfaces using rapid and low-cost 3D printing technology. Several practical applications of the sensor are demonstrated such as the monitoring of the bone-on-bone load and the grasping performance of a biomimetic hand with a tactile sensor printed on the index fingertip. This study can pave the way for the design and fabrication of a new generation of tactile sensors with user-defined auxetic features which can be applied in robotic/prosthetic hands and in pressure/temperature monitoring in impaired human joints. Although I am impressed with the results of the study and have provided some suggestions to improve the quality of the manuscript, I find that the authors have not updated it in a reasonable manner.

Reviewer comment 1a: The sensor can detect both pressure and temperature signals, how can we differentiate these two signals for different purposes?

Reviewer comment 1b: The author only provides a vague response without demonstrating how to differentiate the signals of pressure and temperature. Furthermore:

- 1) Figure S9 shows that at the same pressure level, temperature has a significant influence on the sensor's response.
- 2) This study used a graphene/CNT/silicone rubber composite as a sensing material. It is worth noting that materials based on graphene or CNT are highly sensitive to temperature. Thus, the author's claim ("In contrast, the environmental temperature ranging from 20 to 90°C had negligible effects on pressure sensing based on our experimental results") needs to be further clarified by providing convincing evidence.
- 3) Lastly, in the abstract, the author claimed that "Additionally, the sensor can detect small temperature variations (40 to 90°C)," which further emphasizes the sensor's high sensitivity to temperature.

Reviewer comment 2a: The working mechanism of the tactile sensor needs to be further clarified and explained in a way that is easily understandable for researchers from different fields. Additionally, based on my understanding, this tactile sensor is a resistive-type sensor. Therefore, the terms "first dielectric layer" and "second dielectric layer" used in Figure 1(a) appear to be incorrect since they are conductive. How can they be referred to as "dielectric layers"?

Reviewer comment 2b: The author has not yet corrected the terminology "first dielectric layer" and "second dielectric layer" in Figure 1(a). The author's use of these terms is inaccurate and can lead to misunderstandings.

Reviewer comment 3a: An important parameter for evaluating sensors is the signal-to-noise ratio (SNR). It would be valuable to know the SNR of this tactile sensor and how it compares to other state-of-the-art tactile sensors.

Reviewer comment 3b: The author has not presented any graphs or charts demonstrating the SNR of the sensor. Therefore, it is difficult to determine whether the reported value of "89 dB" is accurate or not, and I remain unconvinced without further evidence.

Reviewer comment 4a: The quality of the figures would benefit from further improvement. For instance, the graphs in Figures 4(a-f) and Figures 7(a-c) appear to be poorly prepared. The black structures in Figure 3 are barely discernible to the naked eye. Additionally, some figures contain excessive details, such as Figure 2.

Reviewer comment 4b: The author has not adequately addressed the issues with the figures. The quality of these figures is still not at a publishable level. It is recommended that the author utilize software like OriginPro to generate more professional-looking graphs.

Based on the aforementioned comments from myself as well as other reviewers, I believe that the authors need to further revise the manuscript and main figures in order to have their work published in Communications Engineering.

Reviewer #4 (Remarks to the Author):

I would like to thank the authors for the effort in addressing the reviewers questions and improving the paper. However, some claims of the authors are not supported by the provided experimental results provided in the paper and the the quality of the presentation and figures (which report still some typos in the labels and refer to first and second dielectric layers as well as too tiny details and poor resolution) is not at the level for this journal.

The answer related to the capability of the sensor to detect shear forces and distinguish between temperature and pressure are not satisfactorily addressed.

The sensor cannot discriminate if a normal pressure with an uneven indenter is applied vs a shear pressure, the tactile image generated can be exactly the same. From sensory-motor control perspective or robot control is important to be able to get information of tangential component of the force but this sensor is not able to do so.

From the temperature - pressure discrimination. The author claims they characterise temperature response and they use different scale depending on the external contact pressure however in operative condition and again with the objective of robot control or sensory motor control, you would need another sensor to detect pressure to be able to decide which scale to use. The validation of this

capability of the sensor has been done in a trivial manner, just applying controlled temperature and looking at sensor output shift however it has not been shown that the sensor can discriminate and provide clear information about contact pressure amplitude and temperature amplitude in the same time.

Reviewer #5 (Remarks to the Author):

After reading the rebuttal letter as well as the revised version of the manuscript, I believe my concerns have been sufficiently addressed by the authors. Thanks to the authors for addressing the points that were mentioned by this reviewer.

However, many figures are still pixelized and not vectorized, which could definitely be improved easily before a formal, final submission. Also, this reviewer couldn't find the revised video file. Only a MEP file, which seems to be a Movavi Video Editor project file could be found. It is probably due to a simple mistake during the submission process. Still, it would be nice to see a final version of the video.

Response to Reviewers' Comments

Paper title: Fully 3D Printed Flexible, Conformal and Multi-directional Tactile Sensor with Integrated Biomimetic and Auxetic Structure

Submitted to: Communications Engineering

Manuscript number: COMMSENG-22-0322A

General Response

We thank the reviewers for their constructive suggestions which have greatly helped us to improve the manuscript. We respond to the individual points in detail below, indicating the quality of the changes that we have made to the manuscript.

Specific Responses to Reviewer #3

- 1. The sensor can detect both pressure and temperature signals, how can we differentiate these two signals for different purposes? The author only provides a vague response without demonstrating how to differentiate the signals of pressure and temperature.*

Response:

Thanks for the comment.

The resistance of the sensor varies with changes in both contact pressure and temperature. It should be noted that the two different signals cannot be effectively differentiated from one another. However, we can adjust the scaling of the pressure signals under different temperatures to mitigate their effects or interference. The effects of temperature response under various contact pressures were studied and analyzed, as depicted in Figure R1. The temperature responses of the sensor were recorded under different contact pressures, demonstrating the sensor's notable sensitivity to temperature variations. To account for different contact pressures, different scales were employed for temperature measurements, necessitating corresponding adjustments to the temperature readings. This information has been added and clarified in lines 407-

425, on page 21-22 of the revised manuscript. The Figure R1 has also been added into the supplementary material (Figure S8). To provide a clearer presentation of the pressure's impact on temperature sensing, we have included more detailed information on the sensor's piezoresistive properties and sensitivities under different pressures in Table S2. This table enables identification of the temperature readings corresponding to various pressures ranging from 0 to 200 kPa.

Furthermore, for the effects of the temperature on pressure sensing, our experimental results indicate that the impact of temperature on the sensor's piezoresistive properties is negligible when the temperature remains below normal room temperature (40°C). The experimental findings reveal that temperature variations ranging from 20 to 40°C have minimal effects on the sensor's performance. Consequently, in most cases, it is unnecessary to scale or adjust the pressure readings when operating within the room temperature range.

In general, based on our experimental results presented in Figure S10 and Table S2, the temperature reading of our sensor can be scaled. However, it is important to note that for most scenarios, the pressure reading under room temperature does not require adjustment. These details have been included in the updated manuscript on lines 345-350 of page 9.

Figure R1. The temperature response of the sensor under different pressures. The sensor was tested under the temperatures ranging from 1 to 200KPa. This data shows that temperature variations within the range of 20 to 40 °C have negligible effects on the sensor's performance, as supported by our experimental results.

2. *Figure S9 shows that at the same pressure level, temperature has a significant influence on the sensor's response.*

Response:

We appreciate the reviewer's observation regarding the influence of temperature on the sensor's response, as depicted in Figure S9. However, there may have been some misunderstanding regarding Figure S9. It actually illustrates the temperature response or temperature sensitivity under different pressures. We have updated Figure S9 to enhance its clarity and improve readers' understanding.

In our study, we conducted experiments to assess and characterize the sensor's temperature sensitivity. By subjecting the sensor to controlled variations in pressure, we were able to observe changes in the sensor's response at different temperatures. Additionally, as mentioned previously, we have included more detailed information on the sensor's piezoresistive properties and sensitivities under different pressures in Table S2. This table facilitates the identification of temperature readings corresponding to

various pressures ranging from 0 to 200 kPa.

However, it is important to note that despite our efforts to quantify the temperature effects, there may still be residual temperature-related influences on the sensor's response. This is a common challenge in many sensing applications, and ongoing research and development efforts are focused on further improving temperature compensation techniques to enhance the sensor's performance in varying temperature conditions.

We acknowledge the significant influence of temperature on the sensor's response, and we have provided relevant data (Table S2) and discussions on Line 407-425 in our revised manuscript to highlight this aspect and the steps we have taken to address it.

3. *This study used a graphene/CNT/silicone rubber composite as a sensing material. It is worth noting that materials based on graphene or CNT are highly sensitive to temperature. Thus, the author's claim ("In contrast, the environmental temperature ranging from 20 to 90°C had negligible effects on pressure sensing based on our experimental results") needs to be further clarified by providing convincing evidence. Lastly, in the abstract, the author claimed that "Additionally, the sensor can detect small temperature variations (40 to 90°C)," which further emphasizes the sensor's high sensitivity to temperature.*

Response:

We appreciate the reviewer's comment and the opportunity to clarify our claim regarding the temperature effects on the pressure sensing performance of our graphene/CNT/silicone rubber composite sensor.

While it is true that graphene and carbon nanotubes (CNTs) can exhibit temperature sensitivity and based on the experimental results shown in Fig. S9, the temperature does have an effect on the piezoresistive properties of the sensor. We apologize for any

confusion caused by the statements "In contrast, the environmental temperature ranging from 20 to 90°C had negligible effects on pressure sensing based on our experimental results" and "the sensor can detect small temperature variations (40 to 90°C). Referring to section 'Characterization of the sensor for temperature sensing and the application on robotics' in the revised manuscript, our experimental results indicate that the temperature effects on the piezoresistive properties of the sensor are negligible when the temperature is below the temperature of 40°C. Figure S9 provides additional data to demonstrate the piezoresistive performance of the sensors under room temperature conditions. The experimental results show that temperature variations ranging from 20 to 40°C have negligible effects on the sensor's performance.

The statements in the 'Results' section have been revised to provide a clear and accurate explanation regarding the negligible effects of temperature variations below room temperature on the pressure sensing performance of our sensor.

The statement in the 'Abstract' section has been modified as follows: " Additionally, the sensor is capable of detecting temperature variations within the range of 40 to 90°C." In our study, while we demonstrate that the sensor is capable of detecting slight temperature variations within this range, it is important to note that our primary focus was on the sensor's pressure sensing capabilities. The revised statement in the abstract ensures that it appropriately reflects the main emphasis of our research on the sensor's pressure sensing performance, while acknowledging its secondary ability to detect small temperature variations.

4. *Reviewer comment 2a: The working mechanism of the tactile sensor needs to be further clarified and explained in a way that is easily understandable for researchers from different fields. Additionally, based on my understanding, this tactile sensor is a resistive-type sensor. Therefore, the terms "first dielectric layer" and "second dielectric layer" used in Figure 1(a) appear to be incorrect since they are conductive. How can they be referred to as "dielectric layers"?*

Response:

Thank you for the comments, and we apologize for the lack of clarity in our manuscript.

Regarding the working mechanism of the tactile sensor, our sensor is based on the piezoresistive effect, which means that the change in resistance of the material is proportional to the applied pressure. The resistive-type sensor consists of two sensing layers (CNT/graphene/silicone composite), and two electrodes (silver-coated copper/silicone composite), which are sandwiched together. When an external pressure is applied, the sensing layer is compressed, resulting in a change in the electrical conductivity of the composite. The change in conductivity is detected by the electrodes and can then be converted into a corresponding pressure signal. The working principle of the tactile sensor has now been further clarified in the caption of Fig.1, and also on Line 115-119, Page 5 of the revised manuscript.

Regarding the terminology used in the manuscript, we apologize for the confusion caused. The "first dielectric layer" and "second dielectric layer" are indeed conductive layers, and we should have named them as "first sensing layer" and "second sensing layer" instead. The purpose of these layers is to provide a conductive path for detecting the changes in electrical conductivity of the sensing layer. In the revised manuscript, we have clearly described the material composition and structure of the sensor, highlighting the conductive layers involved in the pressure sensing process. This will ensure that researchers from different fields can easily understand the working principles of our resistive-type tactile sensor. And the terminologies are all revised accordingly to avoid further confusion,

5. *Reviewer comment 2b: The author has not yet corrected the terminology "first dielectric layer" and "second dielectric layer" in Figure 1(a). The author's use of these terms is inaccurate and can lead to misunderstandings.*

Response:

Many thanks for the comment and we are sorry for the mistake.

The "first dielectric layer" and "second dielectric layer" have now been changed into as "first sensing layer" and "second sensing layer" in Figure 1 (a) to avoid any confusions.

6. *Reviewer comment 3a: An important parameter for evaluating sensors is the signal-to-noise ratio (SNR). It would be valuable to know the SNR of this tactile sensor and how it compares to other state-of-the-art tactile sensors.*

Response:

Thanks for the comment regarding the signal-to-noise ratio (SNR) of our tactile sensor. Evaluating the SNR is indeed important for assessing the performance and quality of sensors.

In our manuscript, we have focused on demonstrating the unique features and capabilities of our tactile sensor, such as its flexibility, sensitivity, and ability to detect both pressure and temperature. While SNR is an important parameter, we did not specifically measure or report the SNR in this study.

To address the reviewer's query and provide a comprehensive evaluation of our sensor's performance, we included an analysis of the SNR in the revised manuscript (Line 270-273, Page 13) and compare the SNR of our sensor with other state-of-the-art tactile sensors reported in the literature. The experimental results and how to calculate the SNR of our sensor have been presented in Fig. S6 in the supplementary material. The SNR is approximately 18.7 dB which is higher/comparable to those piezoresistive pressure sensors in the literature⁵³⁻⁵⁶ ranging from 16 to 80dB, which are due to the optimized material composition and structure design :

53 " Small in-fiber Fabry-Perot low-frequency acoustic pressure sensor with PDMS diaphragm embedded in hollow-core fiber " by Zhao et al. (2018) - SNR of 80 dB

54 " A Universal high accuracy wearable pulse monitoring system via high sensitivity and large linearity graphene pressure sensor " by Jiang et al. (2019) - SNR of 78 dB

- 55 "Piezoresistive Graphene/P(VDF-TrFE) Heterostructure Based Highly Sensitive and Flexible Pressure Sensor" by Kim et al. (2029) - SNR of 60.8 dB
- 56 "High Temperature AlGa_N/Ga_N Membrane Based Pressure Sensors" by Durga et al. (2018) - SNR of 16 dB

This comparison will provide additional insights into the sensor's performance and its suitability for various applications.

7. *Reviewer comment 3b: The author has not presented any graphs or charts demonstrating the SNR of the sensor. Therefore, it is difficult to determine whether the reported value of "19 dB" is accurate or not, and I remain unconvinced without further evidence.*

Response:

Thanks for the comment.

The graph and calculation method for determining the Signal-to-Noise Ratio (SNR) value of 18.7 dB have now been included in Figure S6. The detailed process for calculating the SNR values is provided in the figure's caption. By incorporating this information, we aim to offer readers a clear understanding of how we assessed the SNR of our sensor and how it compares to sensors discussed in the existing literature.

8. *Reviewer comment 4a: The quality of the figures would benefit from further improvement. For instance, the graphs in Figures 4(a-f) and Figures 7(a-c) appear to be poorly prepared. The black structures in Figure 3 are barely discernible to the naked eye. Additionally, some figures contain excessive details, such as Figure 2. The author has not adequately addressed the issues with the figures. The quality of these figures is still not at a publishable level. It is recommended that the author utilize software like OriginPro to generate more professional-looking graphs.*

Response:

Thanks for the comment.

The resolution of Figures 4(a)-4(f) and Figures 7(a)-7(c) has been enhanced and edited using OriginPro to convert them into vectorized diagrams, ensuring high resolution suitable for paper publication.

To provide a clearer presentation of the sensor structures, the printed samples featuring different auxetic structures have been highlighted.

Furthermore, a more detailed description of the experimental process for material preparation and sensor printing has been added to the caption of Figure 2. This addition aims to provide readers with a clear explanation of the process. We have also carefully reviewed and revised the other figures to ensure that important details are clearly visible and excessive details are minimized.

We believe the necessary improvements to the figures have been made to ensure they meet the required publication standards, and the revised figures now better represent the data and findings of our research. We appreciate the reviewer's input, as it has helped us enhance the quality and visual appeal of the figures in our manuscript.

Specific Responses to Reviewer #4

- 1. Some claims of the authors are not supported by the provided experimental results provided in the paper and the quality of the presentation and figures (which report still some typos in the labels and refer to first and second dielectric layers as well as too tiny details and poor resolution) is not at the level for this journal.*

Response:

We appreciate the reviewer's feedback regarding the claims made in our manuscript and the quality of the presentation and figures. We apologize for any inconsistencies and typos in the labeling and descriptions within the paper.

We have carefully revised the manuscript and made the necessary changes to ensure that the claims we make are supported by the experimental results presented in the paper. For example the experimental results and the calculation method for determining the Signal-to-Noise Ratio (SNR) value of 18.7 dB have been included in Figure S6. The detailed process for calculating the SNR values is provided in the figure's caption in supplementary material. By incorporating this information, we aim to offer readers a clear understanding of how we assessed the SNR of our sensor and how it compares to sensors discussed in the existing literature.

Regarding the quality of the figures, we apologize for any difficulties in legibility or resolution. We have taken the reviewer's feedback seriously and have made efforts to improve the clarity and visual appeal of the figures. The resolution of Figures 4(a)-4(f) and Figures 7(a)-7(c) has been enhanced and edited using OriginPro to convert them into vectorized diagrams, ensuring high resolution suitable for paper publication. To provide a clearer presentation of the sensor structures, the printed samples featuring different auxetic structures have been highlighted. Furthermore, a more detailed description of the experimental process for material preparation and sensor printing has been added to the caption of Figure 2 (highlighted in the revised manuscript). This

addition aims to provide readers with a clear explanation of the process. We have also carefully reviewed and revised the other figures to ensure that important details are clearly visible and excessive details are minimized.

We assure the reviewer and the editorial board that we have made the necessary revisions to address the concerns raised. We are committed to providing a high-quality manuscript with clear and well-presented results that meet the standards of this journal. We appreciate the reviewer's valuable feedback, as it has helped us improve the accuracy and presentation of our manuscript.

2. *The answer related to the capability of the sensor to detect shear forces and distinguish between temperature and pressure are not satisfactorily addressed. The sensor cannot discriminate if a normal pressure with an uneven indenter is applied vs a shear pressure, the tactile image generated can be exactly the same. From sensory-motor control perspective or robot control is important to be able to get information of tangential component of the force but this sensor is not able to do so.*

Response:

Thanks for the comments.

The resistance of the sensor varies with changes in both contact pressure and temperature. It should be noted that the two different signals cannot be effectively differentiated from one another. However, we can adjust the scaling of the pressure signals under different temperatures to mitigate their effects or interference. The effects of temperature response under various contact pressures were studied and analyzed, as depicted in Figure R1. The temperature responses of the sensor were recorded under different contact pressures, demonstrating the sensor's notable sensitivity to temperature variations. To account for different contact pressures, different scales were employed for temperature measurements, necessitating corresponding adjustments to the temperature readings. This information has been added and clarified in lines 407-

425, on page 21-22 of the revised manuscript. The Figure R1 has also been added into the supplementary material (Figure S10). To provide a clearer presentation of the pressure's impact on temperature sensing, we have included more detailed information on the sensor's piezoresistive properties and sensitivities under different pressures in Table S2. This table enables identification of the temperature readings corresponding to various pressures ranging from 0 to 200 kPa.

Furthermore, for the effects of the temperature on pressure sensing, our experimental results indicate that the impact of temperature on the sensor's piezoresistive properties is negligible when the temperature remains below normal room temperature (40°C). The experimental findings reveal that temperature variations ranging from 20 to 40°C have minimal effects on the sensor's performance. Consequently, in most cases, it is unnecessary to scale or adjust the pressure readings when operating within the room temperature range.

In general, based on our experimental results presented in Figure S10 and Table S2, the temperature reading of our sensor can be scaled. However, it is important to note that for most scenarios, the pressure reading under room temperature does not require adjustment. These details have been included in the updated manuscript on lines 407-425, on page 21-22.

Figure R1. The temperature response of the sensor under different pressures. The sensor was tested under the temperatures ranging from 1 to 200KPa. This data shows that temperature variations within the range of 20 to 40 °C have negligible effects on the sensor's performance, as supported by our experimental results.

Regarding the shear force sensing, the sensor does respond to stimuli from the transverse direction (shear direction) since the shape and resistance will change due to the interlocked and auxetic structure. The direction of the stimuli can be intuitively differentiated and classified into four principal directions based on each large papilla (highlighted with red squares) surrounded by four small papillae above its four corners. The directions of the shear force can be roughly identified and classified into four principal directions as was shown in Figure R2 below, so the resolution of the shear force differentiation is 90° for this sensor. Figure 2 (g) in the manuscript also shows the shear force applied by fingertip can be detected and the direction can be differentiated. More detailed description on how to classify the stimuli into four orthogonal directions was added to Line 124-131, Page 5-6 of the revised manuscript. The ‘multi-directional’ used in the main text has been replaced by ‘detecting shear stimuli’ instead.

Figure R2. The stimuli are applied from different directions (indicated by the arrows above the sensor) and the corresponding pressure distribution shown through a customized graphical user-interface (GUI). The direction of the stimuli can be intuitively differentiated and classified into four principal directions based on each large papilla (highlighted with red squares) surrounded by four small papillae above its four corners. Therefore, the resolution of the shear force differentiation is 90 degrees for this sensor.

For the case of the uneven indenter stimuli, the use of a small uneven indenter could contribute to a similar pressure distribution as under shear forces. However, for this type of 'uneven indenter' to have protrusions smaller than the sensing element is rare in real-life situations. While the tactile sensor printed on the robotic fingertip could be affected by this circumstance, it is a low-probability event. In most cases, the sensor is in contact with a surface without bumps smaller than the area of the sensing element. It should be noted that this is one of the limitations of our sensor, and it has been mentioned in the discussion on line 490-496, page 24-25 of the revised manuscript. In our research, we have integrated the 3D printed tactile sensor onto a robotic hand, allowing it to accurately perceive shear forces during grasping. By utilizing the sensor's unique capabilities, the robotic hand was able to perform human-like sensorimotor reactions, replicating the intricate interaction between touch and motor control observed in human grasping tasks.

In our study, we have primarily focused on demonstrating the sensor's ability to detect and measure pressure variations. While our sensor does exhibit sensitivity to shear forces, we acknowledge that its current design may not provide a distinct differentiation between normal pressure with an uneven indenter and shear pressure. It is important to note that the ability to precisely distinguish between these two types of forces can be

challenging for many tactile sensors, and it remains an active area of research in the field of tactile sensing. However, we believe that our sensor's unique features, such as its flexibility, high sensitivity, and wide sensing range, still offer valuable applications in various scenarios.

We thank the reviewer for their valuable comments and suggestions, which will help us improve the clarity and accuracy of our manuscript.

3. *From the temperature - pressure discrimination. The author claims they characterize temperature response and they use different scale depending on the external contact pressure however in operative condition and again with the objective of robot control or sensory motor control, you would need another sensor to detect pressure to be able to decide which scale to use. The validation of this capability of the sensor has been done in a trivial manner; just applying controlled temperature and looking at sensor output shift however it has not been shown that the sensor can discriminate and provide clear information about contact pressure amplitude and temperature amplitude in the same time.*

Response:

Thanks for the comment.

We appreciate the reviewer's comment regarding the clarification of the sensor's temperature sensitivity. The 3D printed sensor integrated onto the robotic hand typically operates under room temperature conditions, which are normally below 40 degrees Celsius. It is important to note that while graphene and carbon nanotubes (CNTs) can exhibit temperature sensitivity, our experimental results, as shown in Figure S10, indicate that the temperature effects on the piezoresistive properties of the sensor are negligible at temperatures below the normal room temperature.

We apologize for any confusion caused by our previous statements regarding the temperature effects. To clarify, the statement "In contrast, the environmental temperature ranging from 20 to 90°C had negligible effects on pressure sensing based

on our experimental results" and "the sensor can detect small temperature variations (40 to 90°C)" were intended to emphasize that within the typical operating range of the sensor (below normal room temperature), the temperature effects on the piezoresistive properties of the sensor are negligible based on our experimental results.

To provide further clarity, additional data has been included in Figure S9 to demonstrate the piezoresistive performance of the sensors under room temperature conditions. This data shows that temperature variations within the range of 20 to 40 degrees Celsius have negligible effects on the sensor's performance, as supported by our experimental results.

We appreciate the reviewer's feedback, and we will ensure that these clarifications and additional experimental data are included in the revised manuscript to provide a comprehensive understanding of the sensor's temperature sensitivity and its implications for practical applications. Thank you for bringing these points to our attention, and we value the opportunity to address them in our manuscript.

Specific Responses to Reviewer #5

1. *Many figures are still pixelized and not vectorized, which could definitely be improved easily before a formal, final submission.*

Response:

We appreciate the reviewer's feedback regarding the quality of the figures in our manuscript. We apologize for the pixelation and lack of vectorization in some of the figures. We agree that high-quality figures are essential for effective communication and presentation of scientific results.

The resolution of Figures 4(a)-4(f) and Figures 7(a)-7(c) has been enhanced and edited using OriginPro to convert them into vectorized diagrams, ensuring high resolution suitable for paper publication. We have also carefully reviewed and revised the other figures to ensure that important details are clearly visible and excessive details are minimized. We hope all the necessary improvements to the figures have been made to ensure they meet the required publication standards, and the revised figures now better represent the data and findings of our research.

We appreciate the reviewer's input, as it has helped us enhance the quality and visual appeal of the figures in our manuscript.

2. *Also, this reviewer couldn't find the revised video file. Only a MEP file, which seems to be a Movavi Video Editor project file could be found. It is probably due to a simple mistake during the submission process. Still, it would be nice to see a final version of the video.*

Response:

The authors apologize for the confusion regarding the video file in the revised submission. It appears that there was an error during the submission process, resulting in the incorrect file being included.

To rectify this issue, we have provided a final version of the video for a comprehensive evaluation of our work (Movie.1) in the supplementary materials. The video showcases the experimental setup, the tactile sensor in action, and the sensor's performance in detecting and responding to various stimuli. This will provide a visual demonstration of the capabilities and practical applications of our sensor.

Thank you for bringing this to our attention, and we will make sure to include the correct video file in the final submission for a more comprehensive evaluation of our work.

REVIEWERS' COMMENTS:

Reviewer #4 (Remarks to the Author):

Dear Authors,

I appreciate the effort in revising the manuscript to address the reviewers' concern.

I am happy with the revised manuscript.

Reviewer #5 (Remarks to the Author):

While the video could still be improved, the authors have now addressed most of my concerns.

Response to Reviewers' Comments

Paper title: Fully 3D Printed Flexible, Conformal and Multi-directional Tactile Sensor with Integrated Biomimetic and Auxetic Structure

Submitted to: Communications Engineering

Manuscript number: COMMSENG-22-0322A

General Response

We thank the reviewers for their constructive suggestions which have greatly helped us to improve the manuscript. We respond to the individual points in detail below, indicating the quality of the changes that we have made to the manuscript.

Specific Responses to Reviewer #5

- 1. While the video could still be improved, the authors have now addressed most of my concerns.*

Response:

We appreciate the reviewer's feedback and are pleased to hear that most of their concerns regarding the video have been addressed. We have made efforts to improve the video based on the feedback received, including enhancing its resolution and adding more detailed information to the captions.